# Photo-induced enhancement of lattice fluctuations in metal-halide perovskites

Mingcong Wang [1]✉, Yajun Gao[1], Kai Wang[1], Jiang Liu[1], Stefaan De Wolf [1] & Frédéric Laquai [1]✉

The optoelectronic properties of metal-halide perovskites (MHPs) are affected by lattice fluctuations. Using ultrafast pump-probe spectroscopy, we demonstrate that in state-of-the-art mixed-cation MHPs ultrafast photo-induced bandgap narrowing occurs with a linear to super-linear dependence on the excited carrier density ranging from $10^{17}\,cm^{-3}$ to above $10^{18}\,cm^{-3}$. Time-domain terahertz spectroscopy reveals carrier localization increases with carrier density. Both observations, the anomalous dependence of the bandgap narrowing and the increased carrier localization can be rationalized by photo-induced lattice fluctuations. The magnitude of the photo-induced lattice fluctuations depends on the intrinsic instability of the MHP lattice. Our findings provide insight into ultrafast processes in MHPs following photoexcitation and thus help to develop a concise picture of the ultrafast photophysics of this important class of emerging semiconductors.

[1] King Abdullah University of Science and Technology (KAUST), KAUST Solar Center (KSC), Physical Science and Engineering Division (PSE), Material Science and Engineering Program (MSE), Thuwal 23955-6900, Kingdom of Saudi Arabia. ✉email: mingcong.wang@kaust.edu.sa; frederic.laquai@kaust.edu.sa

Understanding the dynamics of photogenerated carriers in semiconductors has been essential to unleash their maximum performance in optoelectronic applications such as light-emitting diodes and solar cells. In emergent metal-halide perovskite (MHP) semiconductors, Wannier-type excitons are the initial photogenerated species, since the small effective electron mass ($\sim$0.1–0.15 $m_e$)[1,2] and high optical permittivity ($\varepsilon_{opt} \sim$ 4–6.5)[3] yield a Bohr radius larger than the perovskite's lattice constant. Based on the Wannier-exciton approximation, the exciton binding energy ($R_b$) can be estimated to $\sim$54 meV, assuming $\varepsilon_{opt} = 5$ and using the hydrogen model[2]. However, experimentally significantly lower values have been determined, for instance $R_b \sim$ 5–15 meV was found for the classic $CH_3NH_3PbI_3$ (MA) perovskite, based on analysis of the steady-state absorption spectra in the framework of Elliot theory[4], and $R_b \sim$ 10–12 meV was determined independently from the dielectric response from magneto-optical measurements on MA[1], yielding a frequency of 2.44–2.93 THz, which is in the range of optical phonon energies ($\sim$0.5–7 THz)[5]. Thus, the corresponding excitonic dielectric response ($\varepsilon_X \sim 11 \pm 4$) is in the region of optical phonon dielectric responses[6], indicating that the lattice screening is dynamic in nature due to the "soft" crystal lattice of MHPs[7].

Longitudinal optical (LO) phonons dominate the exciton screening, since the measured $R_b$ can be explained when considering strong Fröhlich electron-phonon coupling, resulting from the polar ionic crystal structure of MHPs, which screens efficiently the electron-hole mutual attraction potential[8,9]. Organic cations contribute to the exciton screening via vibrational modes ($\sim$20 meV)[10]. Although cation polarization operates at much lower frequencies ($\sim$100 GHz)[11], they can cause dynamic screening by coupling with the inorganic LO-phonons ($\sim$3–15.8 meV)[12]. Due to efficient thermally-assisted dissociation, excitons are barely observed in MHPs at room $T$ (thermal energy $\sim$25 meV). In fact, ultrashort pump-probe measurements have revealed that the timescale of exciton dephasing is only $\sim$20 fs[13]. Consequently, free carriers are the primary photogenerated species, as also indicated by the quadratic carrier density dependence of spontaneous photoluminescence intensity[14].

However, the transport of photogenerated carriers is limited by localization effects. First, static disorder resulting from thin-film imperfections reduce carrier mobility by scattering[15]. Second, strong Fröhlich coupling drives lattice distortions that tend to slow carriers[16]. The Fröhlich model not only reproduces qualitatively the changes of the carrier mobility depending on different inorganic lattices, but also it yields values only slightly larger than the experimental values[17], suggesting that Fröhlich coupling is the main mechanism limiting carrier mobility in MHPs. However, the Fröhlich model predicts a polaron mobility dependence on temperature of $\sim T^{-0.46}$ (for a range from 200 to 300 K), independent of multi-phonon coupling[18]. This result is inconsistent with the $\sim T^{-3/2}$ dependence observed experimentally[19], which subsequently has been ascribed to large atomic displacements (lattice fluctuations)[11,20]. The thermally-activated lattice fluctuations[21] retard polaron transport by introducing nonlinear-Fröhlich coupling[22], while the random potential fields caused by lattice fluctuations favor quantum Anderson localization[20]. Both are potential mechanisms that can be responsible for the $T^{-3/2}$-dependence.

The energetic barrier for re-orientation and rotation of the organic-cation dipole moments is as small as $\sim$20 meV[23]. Consequently, organic cations can re-orient and rotate around their latticesites at room $T$ (resulting in dynamic disorder), which prolongs carrier lifetimes[24,25] by localizing electrons and holes in different regions[26], implying a tradeoff exists between carrier

lifetime and carrier mobility[27]. The carrier localization induced by dynamic disorder depends also on lattice vibrations. Theoretical studies using density functional theory[28,29] and spectroscopic studies[30,31] have respectively demonstrated that lattice distortions compete with dynamic disorder, leading to the formation of ferroelectric polarons[32].

In this study, we performed transient absorption (TA) pump-probe measurements on thin films of the state-of-the-art triple-cation lead mixed halide perovskite ($Cs_{0.05}FA_{0.8}MA_{0.15}$) $Pb(I_{0.85}Br_{0.15})_3$ or CsFAMA. This perovskite composition has consistently delivered a power conversion efficiency in excess of 20% in conjunction with excellent photostability in perovskite solar cells[33,34]. Carrier density-dependent TA spectra were measured at room $T$ across carrier densities from $N \sim 1.5 \times 10^{17}$ cm$^{-3}$ to $N \sim 3.5 \times 10^{18}$ cm$^{-3}$. We developed a model to analyze the high-energy part of the TA spectra, which revealed weak photo-induced exciton screening and a linear to super-linear dependence of photo-induced bandgap renormalization (BGR) on $N$. This anomalous BGR cannot be explained by carrier-carrier interactions or thermal effects, but can be rationalized by photo-induced lattice fluctuations, which was subsequently supported by time-domain terahertz (td-THz) spectroscopy measurements.

Terahertz spectroscopy is a powerful technique to study photoconductivity[19,35–39]. Fluence-dependent THz photoconductivity spectra obtained from the td-THz spectra show that carrier localization increases with $N$, possibly due to photo-induced lattice fluctuations. We found that the THz photoconductivity spectra present a broadband Lorentz-like resonance, which is governed by size-constrained localization possibly involving Anderson backscattering. The extracted dc carrier mobility ($\omega/2\pi = 0$) decays exponentially with $N$, consistent with a carrier mobility limited by Anderson localization and a Mott mobility edge, indicating photo-induced enhancement of lattice fluctuations. This result is in agreement with the anomalous BGR observed in our TA experiments. The Anderson localization mainly affects the dc carrier mobility ($\omega/2\pi = 0$) at high $N$, while its contribution to the carrier mobility in the THz range ($\omega/2\pi \sim 0.1$–3 THz) at low $N$ is less pronounced, suggesting that the $T^{-3/2}$-dependence of carrier mobility obtained from low-fluence THz experiments is mainly due to nonlinear-Fröhlich coupling. The magnitude of photo-induced lattice fluctuations depends on the intrinsic instability of the MHP lattice. Our findings are related to light-induced fluctuations also observed earlier by other techniques[25,40].

## Results

**Photo-induced transient absorption.** CsFAMA films (thickness $d \sim 300$ nm) were prepared by the solution-processing method on spectroscopic-grade quartz substrates (see methods). Figure 1a shows the steady-state ground-state absorption spectrum, analyzed in the framework of the Elliott model, indicating the absorption (coefficient) consists of a linear combination of excitonic absorption ($\alpha_X$) and absorption from continuum states ($\alpha_C$) as also reported earlier[4,41]. According to the Elliott theory, $\alpha_C$ is enhanced by a factor $\xi$ due to Coulomb attraction between electrons and holes according to[4]:

$$\xi(R_{b0}, x) = \frac{2\pi\sqrt{R_{b0}/x}}{1 - \exp(-2\pi\sqrt{R_{b0}/x})} \quad (1)$$

where $R_{b0}$ is the exciton binding energy, and $x = E - E_g$, where $E_g$ is the band gap. $\xi$ approaches unity when $R_{b0} \to 0$ (L'Hospital's rule), which implies the Coulomb enhancement of $\alpha_C$ does not exist any longer once excitons (electron-hole pairs) are entirely

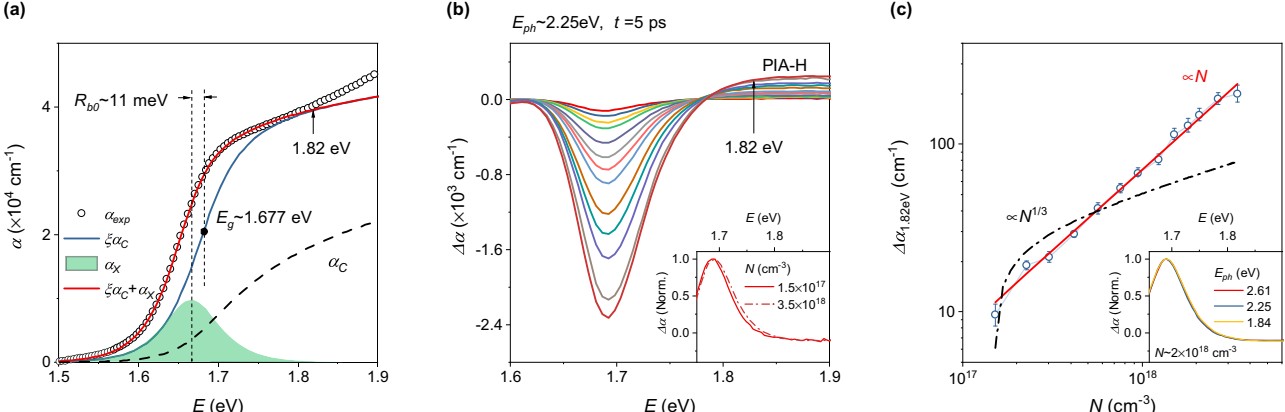

**Fig. 1 Ground-state absorption and transient absorption spectra of CsFAMA thin films. a** Elliott model fit to the ground-state absorption spectrum. The dashed line denotes the continuum absorption, in case excitons are fully screened ($\xi = 1$). The divergence indicated at 1.82 eV implies that parabolic band dispersion is invalid for $E > 1.82$ eV. **b** TA spectra recorded at 5 ps delay time for carrier densities varying from $1.5 \times 10^{17}$ to $3.5 \times 10^{18}$ cm$^{-3}$. The pump wavelength was 550 nm (photon energy ~2.25 eV). PIA-H represents the photo-induced absorption at the high-energy part of the TA spectra. Inset: normalized TA spectra for carrier densities of $1.5 \times 10^{17}$ and $3.5 \times 10^{18}$ cm$^{-3}$, respectively. **c** Plot of the amplitude of the PIA-H signal at $E = 1.82$ eV versus the carrier density. The error bars indicate standard errors. The red solid line is a linear fit to the data. The dash-dotted line is a power law fit with $\propto N^{1/3}$. Inset: normalized TA spectra for a carrier density of ~$2 \times 10^{18}$ cm$^{-3}$, generated by different pump photon energies.

screened. Using $\xi$, the Elliott formula can be expressed also by[4]:

$$\alpha(E) = A \frac{4\pi R_{b0}^{3/2}}{E} \delta(x + R_{b0}) + A \frac{\xi(R_{b0}, x)}{E} \sqrt{x} \qquad (2)$$

where $A$ is a fitting parameter related to the transfer matrix elements, $\delta$ is the Dirac delta function, $\sqrt{x}$ is the normalized density of states in the conduction band assuming a parabolic shape (valid below ~1.82 eV in our sample, see Fig. 1a). The first term accounts for $\alpha_X$, while the second term accounts for $\xi\alpha_C$. To describe the room-temperature absorption spectra, a hyperbolic-secant broadening function accounting for thermal and inhomogeneous broadening was convoluted with Eq. 2 (see SI), yielding $R_{b0} \sim 11$ meV and $E_g \sim 1.677$ eV (Fig. 1a). The decomposed $\alpha_X$ and $\xi\alpha_C$ are plotted in Fig. 1a alongside $\alpha_C$. Clearly, the high-energy part of the absorption spectrum is very sensitive to $\xi$, i.e., the exciton screening.

Next, we focus on the high-energy part of the TA spectra to explore the photo-induced BGR and underlying many-body effects (Fig. 1b). When analyzing the photo-induced absorption at the high-energy part (PIA-H) of the spectra, we can safely neglect $\alpha_X$ (Fig. 1a) and photo-induced broadening (see SI). Therefore, the photo-induced absorption of PIA-H may be described by[41]:

$$\triangle\alpha(E) \cong \frac{A}{E} \xi\left(R_{b0} - \triangle R_b, x - \triangle E_{bgr}\right) \sqrt{x - \triangle E_{bgr}} \left(1 - f_e\right)^2 - \frac{A}{E}\xi\left(R_{b0}, x\right)\sqrt{x} \qquad (3)$$

where $\triangle R_b$ and $\triangle E_{bgr}$ are the screened exciton binding energy and the photo-induced BGR due to many-body effects, respectively. Here, a negative $\triangle E_{bgr}$ represents bandgap narrowing. $f_e = 1/[1 + e^{(E-E_F)/k_B T_e}]$ is the Fermi-Dirac distribution function accounting for the occupation probability of electrons in the conduction band, where $E_F$ is the quasi-Fermi level, $k_B$ is the Boltzmann constant, and $T_e$ is the absolute electron temperature. Since the effective mass of holes is similar to that of electrons[42], we assume that the occupation probability of holes in the valence band is symmetric to that of electrons in the conduction band. For PIA-H recorded 5 ps after photoexcitation, $(1 - f_e)$ is close to 1 due to hot-carrier cooling[43]. In this case, Eq. 3 can be further

simplified (see SI):

$$\triangle\alpha(E) \cong \frac{A}{E} \xi_0 \sqrt{x}\left[\left(\xi_0 e^{-2\pi\sqrt{\frac{R_{b0}}{x}}} - 1\right)\frac{\triangle R_b}{2R_{b0}} - \xi_0 e^{-2\pi\sqrt{\frac{R_{b0}}{x}}}\frac{\triangle E_{bgr}}{2x}\right] \qquad (4)$$

where $\xi_0$ represents $\xi(R_{b0}, x)$ and $\xi_0 e^{-2\pi\sqrt{R_{b0}/x}} \leq 1$.

The photoexcitation was varied across the linear response regime to avoid saturation effects typically occurring at high excitation density (Supplementary Fig. 3). A blue shift of the photo-bleach peak induced by band filling is observed (inset in Fig. 1b), implying that the photo-induced bandgap narrowing should be smaller than the Burstein-Moss shift $\triangle E_{BM}$, which can be estimated from the broadening of the TA photo-bleach peak (~38 meV for $N = 3.5 \times 10^{18}$ cm$^{-3}$)[44]. Hence, an upper limit of $\triangle R_b/R_{b0} \sim 10\%$ can be determined from Eq. (4) using $\triangle E_{bgr} = -38$ meV, indicating that the photo-induced exciton screening is weak. The same consideration was suggested in earlier works when fitting the entire TA spectra[45,46]. In the framework of the hydrogen model, the excitonic dielectric response $\varepsilon_X$ at the ground state can be evaluated by: $R_{b0} = 13.6 \, m_r/(m_e \varepsilon_X^2)$ eV, where 13.6 is the Rydberg constant and $m_r \sim 0.12 m_e$[1,2] is the reduced electron mass, yielding $\varepsilon_X \sim 12.18$. According to the Mott transition criterion of $a_B/\lambda_D = 1.19$[47], where $a_B = 0.0592 m_e \varepsilon_X/m_r$ is the exciton's Bohr radius and $\lambda_D = [\varepsilon_X E_T/(8\pi N q^2)]^{1/2}$ is the Debye screening length, the estimated Mott density in our sample is $N_M \sim 2 \times 10^{16}$ cm$^{-3}$ at room $T$, implying complete exciton screening in the entire excitation regime. The quadratic carrier density dependence of the spontaneous photoluminescence intensity supports the conclusion of complete exciton screening[14]. This indicates that photogenerated excitons in MHPs are dynamic in nature, consequently, they can be monitored by absorption-based measurements (responses to $\varepsilon_X$), but not by photoluminescence measurements (responses to $\varepsilon_s$). We note that the weak exciton screening observed here could indicate the presence of Mahan-type excitons[48]. Mahan-type excitons are known to survive at carrier densities much larger than the nominal $N_M$[49]. However, Mahan-type excitons have only been reported in $CH_3NH_3PbBr_3$ crystals until now, indicated by the observation of strong excitonic peaks even at very high $N$ (~$10^{19}$ cm$^{-3}$)[48].

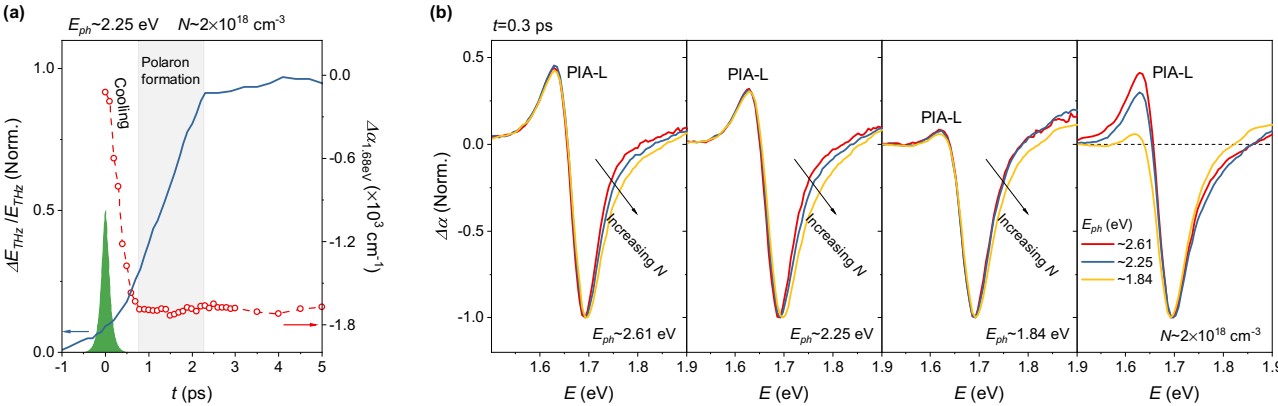

**Fig. 2 The TA spectra of CsFAMA thin films at early times after photoexcitation. a** Left axis: normalized terahertz kinetics (blue solid line). Right axis: time evolution of the TA bleach (peaked at ~1.68 eV) after photoexcitation (red dashed line). The green area represents the correlation (width ~176 fs) of pump and probe laser pulses. The TA bleach reaches its maximum in ~1 ps due to hot-carrier cooling, while the terahertz signal reaches its maximum at ~2 ps. The rise of the terahertz kinetics observed after hot-carrier cooling is an evidence of polaron formation (shaded area). **b** Normalized TA spectra at 0.3 ps delay for different carrier densities and photon energies. PIA-L represents the photo-induced absorption at low probe photon energies, which scales with the carrier density, but decreases with decreasing pump photon energy.

In a doped semiconductor where carrier-carrier interactions are predominant, $\Delta E_{bgr}$ should follow a power law dependence on $N$ according to: $\Delta E_{bgr} = -E_{ex} - E_c \propto N^k$, where $-E_{ex}$ and $-E_c$ are the bandgap narrowing due to exchange correlation and electron-impurity interactions, respectively[50], and $k \sim 1/3$ if electron-impurity interaction is insignificant[51]. However, Fig. 1c shows $k = 1$ for $\Delta\alpha_{1.82eV}$, which appears to be universal in MHPs (Supplementary Fig. 4). This behavior is independent of photon energy, since all TA spectra can be rescaled to similar line shapes (inset in Fig. 1c). Because exciton screening depends either sub-linearly (Wannier-type exciton[48]) or linearly (Mahan-type exciton[49]) on $N$, Eq. (4) suggests $k \geq 1$ for $\Delta E_{bgr}$. Such anomalous BGR cannot be explained by carrier-carrier interactions. Electron-phonon coupling and thermally-induced lattice expansion can be responsible for the observed $k \geq 1$ dependence as they both contribute to the BGR by:[52] $\Delta E_{bgr} = -E'_{ex} - E'_c + E_{ep}$, where $E'_{ex}$ and $E'_c$ account for the exchange correlation and electron-impurity interactions, respectively, if electron-phonon coupling ($E_{ep}$) is included. We believe that photo-induced thermal effects can be neglected since, first, heat accumulation should be negligible, because of the low repetition rate of the pump laser system (3 kHz) used here and thus long off-times between excitation pulses (~333 µs), much longer than the reported heat transport time from the MHP film to the substrate (<9 µs)[53]. Second, the estimated temperature increase of the lattice caused by a single excitation pulse is only about ~0.32 K for $N = 3.5 \times 10^{18}$ cm$^{-3}$ and $E_{ph} = 2.25$ eV (see SI), calculated by using the heat capacity of the MHP lattice (~170–190 J/K$^{-1}$ mol$^{-1}$ at room $T$[54]). Third and importantly, $E_g$ in MHPs increases with the lattice temperature by ~0.3 meV/K[4,55], which reduces the $k$ value rather than increasing it.

Electron-phonon coupling is important in determining the bandgap in MHPs. As mentioned above, unlike classical semiconductors (Si, GaAs, etc.), whose bandgap reduces with temperature, electron-phonon coupling widens the bandgap of MHPs according to[56]:

$$E_{ep}(T) = \int F(\omega, T)\left[n_B(\omega, T) + \frac{1}{2}\right]d\omega \quad (5)$$

where $F(\omega)$ is a spectral function related to the phonon spectra and $n_B$ is the Bose–Einstein distribution of optical phonon modes. Despite $E_{ep}(T)$ being positive, lattice fluctuations can effectively reduce $F(\omega, T)$ and thus introduce a negative component[56],

yielding a temperature-dependent $E_g$ with reduced slope[55]. Thus, the $k \geq 1$ BGR found here is most likely due to an increase of the contribution from photo-induced lattice fluctuations, supported by light-induced fluctuations as revealed by various other techniques[25,40]. The BGR can include a polaronic contribution, since optical spectroscopy measures the bandgap of polarons, so the BGR is possibly linked to polaron formation. However, we have no clear evidence of polaron formation, only indirect evidence is found when comparing the hot-carrier cooling dynamics with the rise of the time-resolved terahertz signal. Figure 2a reveals that the terahertz photoconductivity reaches only half of its maximum, when hot-carrier cooling concludes, which has been ascribed to reduction of the carrier scattering rate due to polaron formation[57].

This scenario can help to clarify why the low-energy photo-induced absorption (PIA-L, Fig. 2b) at $t \sim 0.3$ ps was ascribed to BGR[46] and polaron formation[27], respectively. Figure 2b shows that the amplitude of PIA-L scales with $N$, once again indicating weak photo-induced exciton screening, since a strong exciton screening would significantly reduce the PIA-L amplitude at high $N$. This dependence coincides with the linear dependence of PIA-H on $N$, suggesting that PIA-L is determined by BGR and polaron formation. However, the amplitude of PIA-L increases with the carrier's excess energy (Fig. 2b), which is inconsistent with the dependence of PIA-H on photon energy (inset in Fig. 1c). This discrepancy is likely caused by band filling. Since the PIA-L decay mimics hot-carrier cooling[46], the band filling is predominant in determining PIA-L's amplitude during the hot-carrier cooling process. Therefore, if carriers at the same $N$ (similar $\Delta E_{bgr}$) have different excess energy (significant difference in $T_e$), higher excess energy results in less band filling and thus a larger amplitude of PIA-L.

**Photo-induced THz absorption.** Photo-induced lattice fluctuations affect the carrier photoconductivity in MHPs[12,22]. Measurements of the td-THz spectrum without prior photoexcitation ($E_{THz}$) were performed followed by measurements of the photo-induced td-THz spectra ($\Delta E_{THz}$) recorded at 5 ps after photoexcitation (Supplementary Fig. 7). The differential td-THz spectra in the frequency domain ($\Delta E_{THz}/E_{THz}$) were obtained by Fast Fourier Transformation (Supplementary Fig. 7), and subsequently the photo-induced change of the photoconductivity ($\Delta\sigma$) was

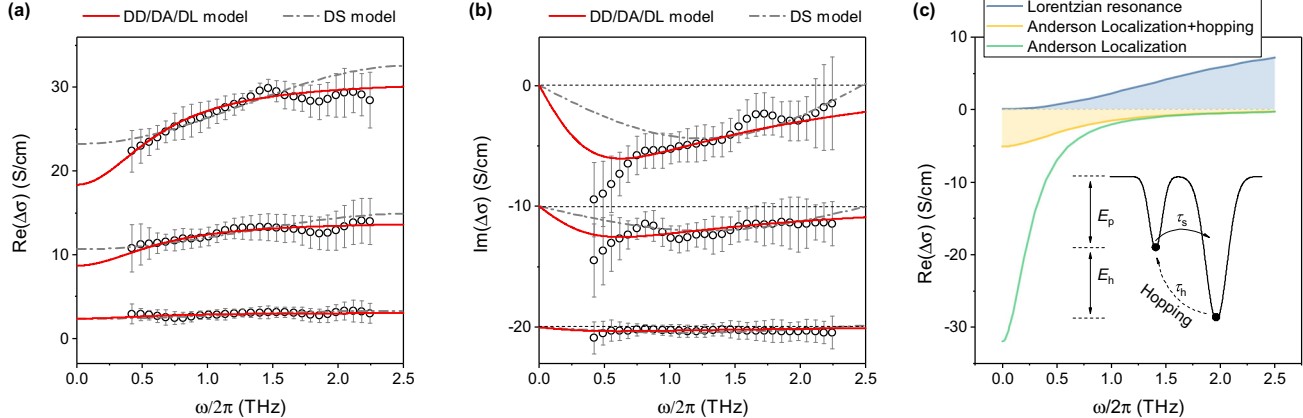

**Fig. 3 Photo-induced changes of terahertz photoconductivity in CsFAMA thin films. a** The real part and **b** the imaginary part of the photo-induced change of the terahertz photoconductivity calculated from the time-domain THz spectra recorded at 5 ps delay time for carrier densities varying from $6.9 \times 10^{17}$ to $7.5 \times 10^{18}$ cm$^{-3}$. The pump wavelength was 550 nm (photon energy ~2.25 eV). All error bars indicate standard errors based on multiple measurements. Dash-dotted lines are fits using the Drude–Smith model (DS model), red solid lines are fits using the Drude–Debye model (DD model). The fits using the Drude–Anderson model (DA model) and Drude–Lorentz model (DL model) are not shown, because they are similar to the DD model fits. **c** Schematic photoconductivity from Lorentzian resonance and Anderson localization. Although Anderson localization substantially reduces the dc photoconductivity, carrier hopping can overcome the Anderson localization, resulting in a photoconductivity response similar to the Lorentzian resonance. Inset: Scheme of carrier localization caused by polaron ($E_p$) formation and due to the presence of an energy barrier ($E_h$).

calculated by[58]:

$$\frac{\triangle E_{THz}}{E_{THz}} = -\frac{1}{\varepsilon_0 c (1 + n_{sub})} \int_0^d \triangle \sigma(\omega, x) dx \qquad (6)$$

where $\varepsilon_0$ is the vacuum dielectric constant, $c$ the speed of light, $n_{sub} \sim 2.13$ the refractive index of the quartz substrate in the terahertz region, $d$ is the sample thickness, and $\omega$ is the angular frequency.

The calculated $\Delta\sigma$ shows a weak ground-state bleach of two transverse-optical phonon modes (Fig. 3a, b, see also Supplementary Fig. 10), indicating the contributions from phononic responses are secondary, consistent with the observations reported in earlier THz works[19,36]. At low $N$, Re($\Delta\sigma$) is virtually frequency independent (Fig. 3a) and Im($\Delta\sigma$) is close to zero (Fig. 3b). These spectral signatures indicate that the photo-conductivity is mediated by free charges that undergo high-rate Drude scattering[38]. At high $N$, Re($\Delta\sigma$) turns downward at the low frequency side and Im($\Delta\sigma$) shows a zero-crossing in the high frequency side. This non-Drude behavior can be assigned to a broadband Lorentz-like resonance centered out of our THz window (Fig. 3c), which is a direct response to the photo-induced lattice fluctuations, or a characteristic of the carrier localization[59]. In fact, the former has been observed in organic crystals[60], supported by the THz lattice fluctuations in MHPs revealed by Raman spectroscopy[61], however, centered at the low-energy side. The latter is common in size-constraint or disordered semiconductors[59].

If the broadband Lorentz-like resonance is a direct response to the photo-induced lattice fluctuations, then the photoconductivity response can be described by the Drude–Debye model (DD model, see SI):

$$\Delta\sigma_{DD}(\omega) = \frac{Nq^2}{m^*} \frac{\tau_s}{1 - i\omega\tau_s} - \sigma_0 \frac{i\omega\tau_r}{1 - i\omega\tau_r} \qquad (7)$$

where $\sigma_0$ is the peak photoconductivity provided by the lattice fluctuations. For experimental THz photoconductivity, $m^* = m_r$ should be constraint in the fit, because td-THz probes both electrons and holes. Clearly, our $\Delta\sigma$ can be well reproduced by the DD model (Fig. 3a, b). The fits yield $\tau_s \sim 1.47$–1.04 fs ($N$-dependent, see Supplementary Table 1 for details),

$\sigma_0 \sim 0.68$–12.29 S/cm ($N$-dependent), $\tau_r \sim 261$ fs (global). The resonance peak can be estimated to ~16–39 meV by $\omega^2_0 = 1/(\tau_r\tau_0)$ using $\tau_s < \tau_0 < 6.37$ fs (see SI). This energy range equals ~3.9–9.5 THz and ~130–390 cm$^{-1}$ and is related to the isolated cation modes[62]. However, these fluctuations are infrared-inactive and thus they cannot be resonant with the THz radiation directly[62].

The photoconductivity response of a localized system is usually described by the Drude–Smith model (DS model) which considers a series of backscattering events[36]:

$$\Delta\sigma_{DS}(\omega) = \frac{Nq^2}{m^*} \frac{\tau_s}{1 - i\omega\tau_s} \left[ 1 + \sum_{j=1}^{\infty} \frac{c_j}{(1 - i\omega\tau_s)^j} \right] \qquad (8)$$

where $q$ is the elementary charge and $\tau_s$ is the carrier's momentum scattering time. $c_j$ is a parameter ($-1 \leq c_j \leq 0$) that is related to the backscattering probability of the $j$th scattering event. Here, the scattering time for each scattering event is considered constant. $c_j = -1$ implies full backscattering (i.e., no conduction), while $c_j = 0$ implies no backscattering (Drude conduction). A negative $c_j$ decreases the dc photoconductivity ($\omega = 0$) by shifting the Drude peak to higher frequency. This approach works well for the case of $c_1 \sim -1$, e.g., isolated nanoparticles with zero dc photoconductivity[63]. In the case of MHPs, their high dc conductivity indicates moderate localization (Fig. 3a), and thus we assume the $j$th scattering event to dominate backscattering. The DS model fits the $\Delta\sigma$ except for Im($\Delta\sigma$) at high $N$ (dash-dotted lines in Fig. 3a, b), which could be caused by the simple assumption that carriers don't change transport direction after any non-backscattering events.

The DS fits yield $\tau_s = 1.76$–1.59 fs ($N$-dependent), $j = 122$ (global), and $c_j = -0.11$–0.18 ($N$-dependent), indicating that backscattering occurs on a timescale of $\tau_b \sim 122\tau_s$, yielding a localization length of $L \sim 3.72$ nm calculated by $L^2 \cong \tau_s\tau_b \sqrt{2E_T/m^*}$[64] using $\tau_s = 1.76$ fs, $E_T = 25$ meV, $m^* = 2m_r \sim 0.24$ $m_e$. Comparison of the film thickness ($d \sim 300$ nm) and grain size (>300 nm) with $L$ yields $d/L \sim 80$. The results of $c_j = -0.11$–0.18 and $d/L \sim 80$ are consistent with the previous assumption of moderate size-constrained localization. The likely

explanation for $j = 122$ is the quantum Anderson localization[20]. Due to the random electrostatic potential induced by lattice fluctuations, Anderson backscattering can occur after many scattering events with a backscattering time $\tau_b \gg \tau_s$[20]. Since Anderson backscattering results in substantial reduction of the dc photoconductivity (Fig. 3c), the dc photoconductivity of localized states is governed by carrier hopping (Fig. 3c), which breaks the Anderson localization in the dephasing time $\tau_d$[20]. The Drude–Anderson model (DA model) is given by[64]:

$$\Delta\sigma_{DA}(\omega) = \frac{Nq^2}{m^*}\left[\frac{\tau_s}{1 - i\omega\tau_s} - \frac{\phi\phi_b\tau_s}{1 - i\omega\phi_b\tau_b}\right], \phi_b = \frac{\tau_d}{\tau_d + \tau_b} \quad (9)$$

where $0 \le \phi \le 1$ is the fraction of localized carriers[20] and $0 < \phi_b \le 1$ is a parameter accounting for the hopping dephasing. Here, $\phi\phi_b$ is the effective fraction of localized carriers and $\phi_b\tau_b$ is the effective backscattering time. The DA fits the experimental $\Delta\sigma$ similarly well as the DD model, since they are mathematically equivalent (see SI). The fits yield $\tau_s \sim 1.9$–$1.76$ fs ($N$-dependent), $\phi\phi_b \sim 0.23$–$0.42$ ($N$-dependent), and $\phi_b\tau_b \sim 245.6$ fs (global).

Both the DS model and DA model are backscattering models that cannot describe the case of localization due to an energy barrier characterized by a barrier height ($E_h$) and hopping time ($\tau_h$) (scheme in Fig. 3). The thermally-assisted hopping time can be expressed as $\tau_h = \tau_s \exp(E_h/k_BT)$. When $E_h \to 0$, $\tau_h$ reduces to the Drude scattering time $\tau_s$. In this case, the photoconductivity response can be expressed by the Drude–Lorentz model (DL model):

$$\Delta\sigma_{DL}(\omega) = \frac{Nq^2}{m^*}\left[\frac{(1-\phi)\tau_s}{1 - i\omega\tau_s} + \frac{\phi\omega\tau_h}{\omega - i\tau_h(\omega^2 - \omega_h^2)}\right], \omega_h = \frac{E_h}{\hbar} \quad (10)$$

where $0 \le \phi \le 1$ is the fraction of localized carriers. Since $\omega\tau_s \ll 1$ is satisfied in the THz window, the DL model becomes mathematically equivalent to the DA model, when $\phi = \phi\phi_b$, $\omega_h^2 = 1/(\tau_s\phi_b\tau_b)$, and $\tau_h = \tau_s$ (see SI) however, the two models have different physical implications. The DL fits are similar to the DD fits in Fig. 3a, b except, they yield $\tau_s \sim 1.68$–$1.31$ fs, $\phi \sim 0.1$–$0.22$, $E_h \sim 22.6$ meV. The localized pairs with energy $E_p + E_h$ are usually the surface/interface plasmons caused by carrier accumulation at grain boundaries and the film surface[65]. However, plasmon resonance can be ruled out, because its peak position is proportional to $\sqrt{N}$, yet no obvious peak shift can be observed here when $N$ increases (Fig. 3a). Therefore, we hypothesize that the broadband Lorentz-like resonance is caused by size-constrained localization due to Anderson backscattering.

## Discussion

Based on the finding of Anderson localization, Fig. 4a shows the Drude mobility ($\mu_s$) calculated from the momentum scattering time ($\mu_s = e\tau_s/m^*$), together with the carrier mobility calculated from the frequency-averaged photoconductivity ($\mu_{avg} = \sigma_{avg}/eN$), and the dc mobility calculated by $\mu_{dc} = (1 - \phi\phi_b)\mu_s$. The decrease of $\mu_s$ with $N$ may be related to the photo-enhanced nonlinear-Fröhlich coupling introduced by lattice fluctuations[12,22]. $\mu_{dc}$ is smaller than $\mu_s$ due to the Anderson localization. Interestingly, $\mu_{dc}$ decreases faster than $\mu_s$, since $1 - \phi\phi_b$ decreases with $N$ (Fig. 4b), indicating photo-enhanced Anderson localization and thus lattice fluctuations, consistent with the conclusion drawn from $\mu_s$. We note that $\mu_{avg}$ is higher than $\mu_{dc}$ because it partially accounts for the non-dc mobility contributed by localized carriers. However, $\mu_{avg}$ has a similar $N$-dependence as $\mu_s$ and unlike $\mu_{dc}$, indicating that $\mu_{avg}$ is dominated by $\mu_s$. While Anderson localization results in a carrier mobility obeying the experimentally observed $T^{-3/2}$-dependence[20], this finding suggests that the $T^{-3/2}$-dependence of

the carrier mobility determined from $\sigma_{avg}$ is primarily caused by nonlinear-Fröhlich coupling, especially for low-fluence measurements[19,38]. It is noteworthy that the carrier mobility obtained from microwave photoconductivity may be mainly determined by Anderson localization, because microwave is closer to $\omega = 0$[66].

The DA model also fits the data for MAPbI$_3$ (MA) and FA$_{0.83}$MA$_{0.17}$PbI$_{2.49}$Br$_{0.51}$ (FAMA) (Supplementary Fig. 11). The ratio of $\mu_{dc}/\mu_s = 1 - \phi\phi_b$ decreases exponentially with $N$ (Fig. 4b). According to the Mott mobility edge theory[67], $\mu_{dc}/\mu_s$ decreases exponentially according to:

$$\frac{\mu_{dc}}{\mu_s} = \exp\left(-\frac{E_M - E_F}{k_BT}\right) \quad (11)$$

where $E_M$ is the mobility edge determined by lattice fluctuations[67]. Since after hot-carrier cooling (i.e., $t = 5$ ps), $E_F$ and $T$ depend only weakly on $N$, the exponential decay of $1 - \phi\phi_b$ results in an increase of $E_M$ and thus an increase of lattice fluctuations, consistent with the conclusion drawn from the $k \ge 1$ dependence of the BGR observed in the TA spectra. Consequently, the product $\phi\phi_b$ is a measure of lattice fluctuations. We note that $\phi\phi_b$ at $N = 0$ is nonzero due to the presence of thermally-activated lattice fluctuations. The large value of $\phi\phi_b$ ($\sim 0.46$) in FAMA indicates that FAMA is intrinsically unstable, possibly due to the larger cation size of FA$^+$, and/or the dynamic disorder of FA$^+$ re-orientation and rotation. However, incorporating only 5% Cs$^+$ into the FAMA structure greatly improves the lattice stability, as evidenced by the small background of $\phi\phi_b$ ($\sim 0.21$) in CsFAMA. This improvement has previously been ascribed to the smaller size of Cs$^+$ which results in a lattice structure closer to the Goldschmidt tolerance range[68], indicating that the contribution of dynamic disorder from FA$^+$ to the lattice fluctuations is secondary.

By comparing the exciton binding energy $R_{b0}$ with the Drude scattering time $\tau_s$ (Fig. 4c), we find that $\tau_s$ has a positive correlation with $R_{b0}$, which appears reasonable since both $\tau_s$ and $R_{b0}$ are affected by electron-phonon coupling. However, this is no longer valid if the unscreened exciton binding energy is very different[9]. $\tau_s$ of all samples decreases with the population of excited states, as discussed above, which can be a result of the photo-enhanced nonlinear-Fröhlich coupling introduced by lattice fluctuations[12,22]. This is supported by the larger $\phi\phi_b$ for excited states and the anti-correlation between $\tau_s$ and $\phi\phi_b$ (Fig. 4c). In fact, the same behavior was proposed theoretically[69] and experimentally observed in PL spectra[70], but it has been ascribed to the polaron destabilization related to the wave function overlap between neighboring polarons[70]. The photo-induced lattice fluctuations reported here could be related to light-induced fluctuations observed by other techniques[25,40], with magnitudes depending on the intrinsic instability of the MHP lattice (Fig. 4c). The Anderson localization scenario suggests coexistence of band-like and hopping transport which challenges the standard band-like carrier transport model for perovskites[24]. However, the precise nature of carrier transport in MHPs remains an open question. Here, we emphasize that the observation of photo-induced lattice fluctuations, which is the main finding of our work, is independent of the Anderson localization. In fact, band-like transport models (Supplementary Tables 1–3) indicate a photo-induced increase of the dielectric response (DD model) or carrier localization (DS model and DL model). To some extent, both can be related to photo-induced lattice fluctuations.

In conclusion, triple-cation MHP films (CsFAMA) were investigated by TA and td-THz spectroscopy. We developed a model to analyze the high-energy part of the TA spectra, which takes into account photo-induced exciton screening and

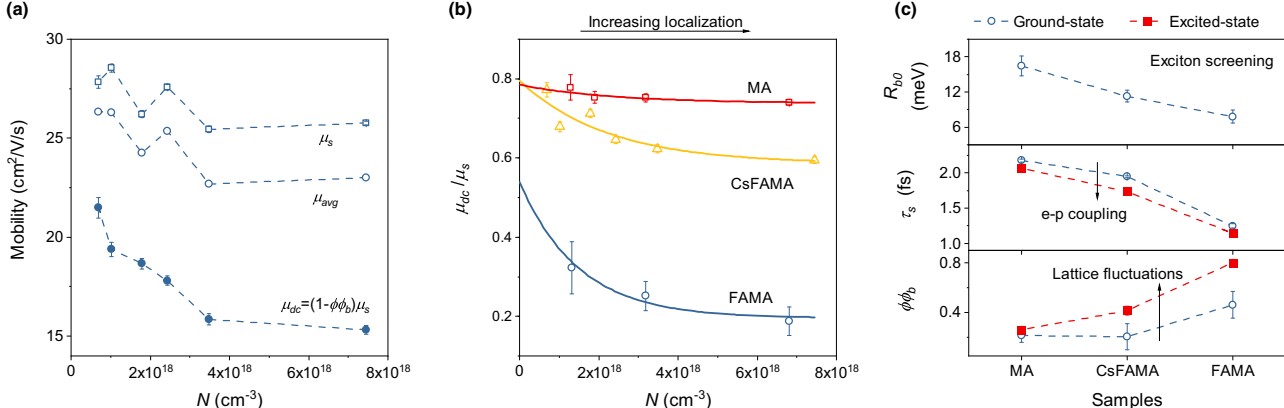

**Fig. 4 Photo-induced carrier localization. a** The carrier mobility ($\mu_s$, open squares) obtained from the fitted value of the momentum scattering time ($\tau_s$). $\mu_{avg}$ is the carrier mobility calculated from the frequency-averaged photoconductivity (open circles). $\mu_{dc}$ represents the dc mobility (solid circles). **b** The ratio of $\mu_{dc}/\mu_s = 1 - \phi\phi_b$ decreases with carrier density, indicating photo-induced carrier localization. Solid lines are exponential fits to the data of MA, FAMA and CsFAMA thin films, respectively. **c** Comparison of $\tau_s$ and $\phi\phi_b$ between ground-state and excited-state together with the ground-state exciton binding energy ($R_{b0}$). The black arrows indicate photo-induced enhancement of electron-phonon coupling and lattice fluctuations. All error bars indicate a 95% confidence interval.

bandgap narrowing. The bandgap narrowing was found to increase linearly or super-linearly with carrier density, which cannot be explained by carrier-carrier interactions, but can be rationalized by photo-induced lattice fluctuations, as supported by the increased carrier localization in the terahertz photoconductivity spectra. Anderson backscattering is the most likely mechanism responsible for the increased carrier localization. The photo-induced lattice fluctuations aggravate the randomness of the quantum electrostatic potential and thus increase carrier localization. Finally, the comparison of Anderson localization between different MHP systems revealed that the magnitude of photo-induced lattice fluctuations correlates with intrinsic instability of the MHP lattice. Our findings provide specific insight into the excited-state photophysics of state-of-the-art MHPs, and they aid the development of a concise picture of the ultrafast physics of this important class of semiconductors.

## Methods

Materials: $PbI_2$ and $PbBr_2$ were purchased from TCI. FAI and MABr were purchased from Dyesol. CsI, RbI and all anhydrous solvents (DMF, DMSO, chlorobenzene) were purchased from Sigma-Aldrich. $SnO_2$ colloid precursor was obtained from Alfa Aesar, the particles were diluted by $H_2O$ and isopropanol to 2.67%. All chemicals were used without further purification.

Perovskite film fabrication: $MAPbI_3$ perovskite precursor solution in a mixed solvent (DMF/DMSO = 9:1) was used. In total, 70 µl of perovskite solution was spun onto the substrates at 4000 rpm for 30 s. In total, 150 µl of chlorobenzene was dropped in the center of the substrates 22 s before the end of the spin-coating process. After the rotation ceased, the substrates were immediately transferred onto a hotplate of 100 °C and annealed for 10 min. $PbI_2$ (508 mg, 1.1 mmol), $PbBr_2$ (80.7 mg, 0.22 mmol), FAI (171.97 mg, 1 mmol) and MABr (22.4 mg, 0.2 mmol) in 1 ml of a 4:1 (v/v) mixture of anhydrous DMF and DMSO. This resulting precursor solution for $(FA_{0.83}MA_{0.17})Pb(I_{0.83}Br_{0.17})_3$ contains a 10 mol% excess of $PbI_2$ and $PbBr_2$, respectively, which was introduced to enhance device performance. The FAMA solution was filtrated through a 0.45 µm syringe filter before use. CsI was dissolved in 1 ml DMSO and 42 µl of the ~1.5 M CsI stock solution was added to 1 ml FAMA solution to get $(Cs_{0.05}FA_{0.8}MA_{0.15})Pb(I_{0.85}Br_{0.15})_3$ solution. In total, 70 µl of perovskite solution was spun onto the substrates at 2000 rpm for 10 s and 4000 rpm for 30 s. In total, 150 µl of chlorobenzene was dropped in the center of the substrates 10 s before the end of the spin-coating process. After the rotation ceased, the substrates were immediately transferred onto a hotplate of 100 °C and annealed for 10 min.

TA setup: our TA setup uses a commercial Ti:sapphire amplifier operating at 800 nm with a repetition rate of 3 kHz as laser source. Its pulse width (FWHM) is compressed to ~125 fs. Two optical parametric amplifiers are used to tune the laser wavelength. The white-light probe is generated by 1300 nm laser (from TOPAS1) with a $CaF_2$ crystal that mounted on a continuously moving stage, which enables us to generate a super-continuum pulses with a spectral range from 350 to 1100 nm. The pump laser (from TOPAS2) is chopped to 1.5 KHz and delayed by an automated mechanical delay stage (Newport linear stage IMS600CCHA) from −400 ps to 8 ns. Pump and probe beams were overlapped on the front surface of

the sample, and their spot sizes (D86 ~ 3 mm) were measured by a beam viewer (Coherent, LaserCam-HR II) to make sure the pump beam was about three times larger than the probe beam (D86 ~ 1 mm). The perovskite samples are stored in a nitrogen-filled chamber to protect from degradation, and photo-excited by 475, 550, and 675 nm in this work. The probe beam was guided to a custom-made prism spectrograph (Entwicklungsbüro Stresing) where it was dispersed by a prism onto a 512 pixel complementary metal-oxide semiconductor linear image sensor (Hamamatsu G11608-512DA). In order to account for the reflection, we first measured the transient reflection ($\Delta R/R$), then measured the TA ($\Delta T/T$).

td-THz setup: our td-THz setup uses the same Ti:sapphire amplifier as the TA setup. The THz emitter and detector are two 1 mm thick <110> oriented zinc telluride (ZnTe) crystals. All the THz related optics were placed in a closed chamber, which was continuously purged with pure nitrogen gas. Perovskite samples were excited by 550 nm laser pulses obtained from the same TOPAS2 as used in the TA experiment. A motor equipped with a circular ND filter was used to change the pump fluence in the fluence-dependent experiments.

## Data availability

The main data supporting the findings of this study are available within the Article and its Supplementary Information. Extra data are available from the corresponding author (M.W. or F.L.) upon request.

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

## Acknowledgements

This publication is based upon work supported by the King Abdullah University of Science and Technology (KAUST) Office of Sponsored Research (OSR) under Award No: OSR-2018-CARF/CCF-3079, OSR-2018-CRG7-3737, OSR-2019-CRG8-4093, and OSR-2020-CRG9-4350.

## Author contributions

M.W. performed the time-domain terahertz measurements and data analysis. He conceived and developed the theoretical model and wrote the initial draft of the paper. M.W. and Y.G. performed the ground-state absorption and transient absorption measurements. K.W. and J.L. prepared the perovskite thin-film samples. S.D.W. and F.L. supervised the work and revised the manuscript. All authors contributed to refining the final version of the manuscript.

## Competing interests

The authors declare no competing interests.
