## [Peer Review File · Nature Communications]

Title: Photo-induced enhancement of lattice fluctuations in metal-halide perovskitesREVIEWER COMMENTS

Reviewer #1 (Remarks to the Author):

This paper reports on a combination of ultrafast transient absorption and time-resolved THz spectroscopy measurements on the triple cation metal halide perovskite (FAMAC) over a wide carrier density range at room temperature. The authors find transient band gap narrowing by careful analysis of the transient absorption spectra and application of Elliott theory. Transient THz spectra shows a marked reduction in conductivity at low frequencies and this is assigned to a Debye response arising from a photo-induced modulation of a lattice-related dipole. Analysis of these contributions leads the authors to claim a photoinduced increase in electron-phonon coupling which drives polaron formation. They also extend these measurements to deduce a critical density of polarons where lattice strain is reduced due to overlap of adjacent polarons.

Overall I found the paper quite interesting, of broad interest and well written. However I have several questions and comments:

In the introduction, the authors revive an old misconception in the literature that when discussing the exciton binding energy one should employ the high frequency dielectric constant when talking about screening of the Coulomb interaction. There is some ambiguity that plagues the literature as to which limit to take. The rule of thumb is if $R^*/\hbar > \omega_{TO}$, the high frequency limit is appropriate. If $R^*/\hbar < \omega_{TO}$, the static limit is more appropriate to describe the screening. The authors assume that $R^*/\hbar > \omega_{TO}$ when using ϵ_{∞} , which then gives a Ry energy of 50 meV, larger than w_{TO} . So it is self-fulfilling. If you had used the static permittivity, you would calculated an R^* 6 times lower and again it would look correct. There are a few measurements that are Independent of this assumption at least on MAPI (e.g. high field magnetoabsorption measurements by Miyata and Zhu) and I suggest using these instead, which determine Ry^* to be on the order of 10 meV. This is comparable to w_{TO} and suggests that the lattice screening of the exciton is dynamic in nature, so it's probably not so simple.

The spectral resolution for the THz measurements in Fig. 3 looks heavily interpolated due to zero-padding the Fourier transform, which effectively Gaussian blurs each data point. This might wash over features in the spectrum due to the strong polar optical phonons that may be present in the system. The effect of this may be to assign phonon features to mobile carriers. The authors should replot their data without such over-padding to show the level of the noise and whether or not there are spectral features that might be due to optical phonons in the film. These phonon features have been seen in other THz studies on films.

Can the authors be a little more clear as to what the dipolar relaxation is the Debye model fitting? The text seems to suggest that it is related to the lattice, so is it a photoinduced relaxation of the cation orientation? It's hard to tell whether it's physically realistic or just that the model fits.

For the THz spectroscopy, how large was the pump beam spot at the sample? If the pump beam is too small compared to the frequency dependent THz spot size, there is a suppression in THz conductivity at low frequencies that looks exactly like their data. Given that they are pumping with the output of a TOPAS and this output is typically only 1mm or less, this is a serious possibility. See for example Fig. 2 in the paper: M. C. Beard, G. M. Turner and C. A. Schmuttenmaer, Phys. Rev. B 62, 15764 (2000) where they show a strong suppression for a pump spot of 1.1 mm diameter. The authors should confirm that their data shows no such dependence on the pump spot size and it is truly a homogeneous excitation for all frequencies in the THz pulse bandwidth. Did the authors check that a Drude model was observed in the same experimental conduction using a typical Drude semiconductor like GaAs, for instance?

The data should have some indication of measurement uncertainty. This could easily be done by presenting the mean and the standard error of the mean obtained from analysis of the multiple time-domain waveforms. The fits should then take into account the measurement uncertainty, which will be greater at the lowest and highest of their frequency window. There is quite a bit of smoothing in the data.

The authors make a connection between the change in the Debye contribution and the electron-phonon coupling strength using a polaron mobility model, taking into account the expected increase in the effective mass and therefore mobility upon dressing with LO phonons. They find the fitted static dielectric permittivity is less than the theoretically predicted value, however has this not been measured? I'm also confused as to why they say that optical phonons contribute weakly to the dielectric response. These are highly polar materials and the optical phonons contribute significantly to the dielectric response. Perhaps they meant the LO phonons? Even then the Lyddane-Sachs-Teller relation connects the LO phonon frequency to the static permittivity, so one can estimate the contributions.

More on the editorial side, there appears to be something wrong with the bibliography. The parameters related to the polaron formation seem to be taken from Ref. 39, however no reference exists in the text. Instead the authors refer to Ref 36 on line 203 which concerns exciton-Mott physics unrelated to perovskites. Reference 40 is referred to when discussing polaron formation (line 205) however I believe this should be Refs 39 that measures the polaron formation in MAPI and 41 which investigates the frequency averaged risetime with excess energy and assigns it to polaron formation (mentioned in line 211).

Reviewer #2 (Remarks to the Author):

In their manuscript, Wang et al present an experimental study on the ultrafast relaxation of lead halide perovskite. Using a combination of transient absorption and time domain terahertz spectroscopy, they study the band-gap narrowing following an optical excitation, and claim that their observations are uniquely explained by a photo-induced, transient increase in electron-phonon coupling. While the authors results are interesting, I find the analysis not entirely convincing. Mostly, this is due to

implications that seem inconsistent with other observations in the literature and an insufficient discussion of the underlying mechanism. As such, I believe publication at this time would be premature.

The authors might consider the following as they revise their manuscript:

There is very little discussion of temperature induced broadening in the manuscript and no attempt to rule it out as the reason for the observed band gap narrowing. First, the static broadening contribution is not discussed sufficiently. The SI states the functional form used, but not the parameters nor a study their sensitivity. Is σ changed in the excited state? Further, the excitation is significantly above the band gap and thus there is a significant amount of energy that must be dissipated by the lattice. If polarons are formed, there must be a large energy transfer to the lattice. However, perovskites are notoriously poor thermal conductors. Can local heating explain the band gap renormalization (either directly, or indirectly due to local lattice expansion and accompanying strain)?

Many estimates of polaron binding energies are $\sim kT$ [1,2], which are consistent with the present authors study (line 204). However such a weak binding would not result in a significant population of localized carriers. Indeed, many other estimates of the number of free carriers report it to be much higher under these excitation densities [3]. How do the authors justify this discrepancy, both conceptually with the low binding energy and experimentally with the inconsistency with previous reports.

1. Bischak, Connor G., et al. "Tunable polaron distortions control the extent of halide demixing in lead halide perovskites." *The journal of physical chemistry letters* 9.14 (2018): 3998-4005.

2. Frost, Jarvist Moore. "Calculating polaron mobility in halide perovskites." *Physical Review B* 96.19 (2017): 195202.

3. Wehrenfennig, et al. *Advanced materials* (2014)

Finally, there is no attempt to associate a mechanism to the increased electron-phonon coupling. Typically, the static dielectric is decreased with increased carriers as polarization fluctuations (which determine the dielectric constant) become constrained. To have a photo-induced effect that was not just from heating, the authors would have to posit the emergence of a novel motion enabled under excitation. Further, the authors discussion is limited to a simple model of electron phonon coupling. A significant amount of work has gone into the study of this coupling, demonstrating that it is not well described by a Frohlich model [4,5] The authors would benefit from considering, and ruling out, contributions not contained in this model.

4. Mayers, Matthew Z., et al. "How lattice and charge fluctuations control carrier dynamics in halide perovskites." *Nano letters* 18.12 (2018): 8041-8046.

5. Schlipf, Martin, Samuel Poncé, and Feliciano Giustino. "Carrier lifetimes and polaronic mass enhancement in the hybrid halide perovskite $\text{CH}_3\text{NH}_3\text{PbI}_3$ from multiphonon Fröhlich coupling."

Physical review letters 121.8 (2018): 086402.

Reviewer #3 (Remarks to the Author):

In this paper the authors report photo-induced enhancement of electron-phonon coupling and dynamic disorder in triple cation mixed halide perovskite using ultrafast transient absorption and THz spectroscopies. The authors' conclusions are based on observation of linear dependence of photoinduced band-gap normalization on the carrier density and Debye relaxation component observed in the terahertz photoconductivity spectra. This paper reports results of great general interest. Unfortunately, it overlooks several results that are not explained and commented upon. The authors present the portion of the data and fittings. I would find it clearer for the authors to first present all the data and fitting curves (with error estimations) so that the reader gets a clear view of the interpretation and analysis. My concerns and suggestions are detailed below.

1) While describing the high-energy tails in transient absorption spectra, authors have mentioned that "Rb=Rb0 was used in the fitting, since even a small reduction in Rb resulted in poor fits" (Page 7, line 144). This fitting has a huge importance to the authors' conclusions as it directly relates to the red-shifted excitonic level. So it is essential that the authors mention the parameter space tried for fitting along with the error estimations.

2) Authors have mentioned that "For $N \sim 4.3 \times 10^{18} \text{ cm}^{-3}$, ΔE_{bgr} assumes a plateau between ~ 2 -5 ps after reaching the maximum (Fig. 2a), indicating photo-induced BGR is concluded in ~ 2 ps" (Page 9, line 170). With all due respect, I don't see that in Fig. 2a. ΔE_{bgr} seems to maximize at ~ 700 -800 fs and then shows a decay followed by ill-defined dynamics in 2-5 ps region, which is defined as plateau by the authors. Similar plateau is also observed in 10-30 ps region. Can authors clarify this? Also, what is the cause of decay of ΔE_{bgr} after reaching the maximum? Can it be contributed by carrier-carrier interactions?

3) It is important that the authors show the time-dependent BGR for other carrier densities in supporting information. It might also help the authors to answer questions raised by me in point 2.

4) Authors have also performed transient absorption measurements on FAMARb and FAMA. Again, it will be useful to present time-dependent BGR data for these systems as well. Does the Rb=Rb0 fitting also holds for these systems (point 1). Please show the fitted data in the supporting information. It is very unfortunate that the data on these two systems have not been given their due importance. In fact, it has only been mentioned once in the main manuscript. Also, authors have not provided the rationale behind probing these specific systems.

5) Authors have used different pump wavelengths for transient absorption (2.61 eV) and tdTHz (2.25 eV). Can authors please explain the reason for this? It might result into different dynamics and coupling to different phonons at different excitation frequencies.

6) Can authors provide tdTHz data on other two perovskite systems as well? This will really strengthen the inferences drawn using combination of the classical Drude model and the Debye relaxation model to describe fluence-dependent THz photoconductivity spectra.

Minor points:

7) Main manuscript and Supplementary information have different titles.

8) Page 6, line 123: carrier density should be 3.6×10^{16} and not 3.6×10^{18} .

9) Page 2, line 37: excitons are observed at room temperature. Refs: Nature Communications volume 11, Article number: 850 (2020); ACS Photonics 2018, 5, 3, 852–860.

10) Figure 3a inset is hard to comprehend.

11) Authors might also consider to refer following papers at appropriate places in the introduction: Science Advances 2019, Vol. 5, no. 5, eaaw5558; Journal of Applied Physics 124, 215106 (2018).

Reviewer #1 (Remarks to the Author):

This paper reports on a combination of ultrafast transient absorption and time-resolved THz spectroscopy measurements on the triple cation metal halide perovskite (FAMAC) over a wide carrier density range at room temperature. The authors find transient band gap narrowing by careful analysis of the transient absorption spectra and application of Elliott theory. Transient THz spectra shows a marked reduction in conductivity at low frequencies and this is assigned to a Debye response arising from a photo-induced modulation of a lattice-related dipole. Analysis of these contributions leads the authors to claim a photoinduced increase in electron-phonon coupling which drives polaron formation. They also extend these measurements to deduce a critical density of polarons where lattice strain is reduced due to overlap of adjacent polarons.

Overall I found the paper quite interesting, of broad interest and well written. However I have several questions and comments:

Authors' response: We thank the reviewer for his/her encouraging comments on our work. Please find below our point-by-point response to the comments.

In the introduction, the authors revive an old misconception in the literature that when discussing the exciton binding energy one should employ the high frequency dielectric constant when talking about screening of the Coulomb interaction. There is some ambiguity that plagues the literature as to which limit to take. The rule of thumb is if $R^*/\hbar > \omega_{TO}$, the high frequency limit is appropriate. If $R^*/\hbar < \omega_{TO}$, the static limit is more appropriate to describe the screening. The authors assume that $R^*/\hbar > \omega_{TO}$ when using ϵ_{∞} , which then gives a Ry energy of 50 meV, larger than ω_{TO} . So it is self-fulfilling. If you had used the static permittivity, you would calculate an R^* 6 times lower and again it would look correct. There are a few measurements that are independent of this assumption at least on MAPI (e.g. high field magnetoabsorption measurements by Miyata and Zhu) and I suggest using these instead, which determine Ry^* to be on the order of 10 meV. This is comparable to ω_{TO} and suggests that the lattice screening of the exciton is dynamic in nature, so it's probably not so simple.

Authors' response: We agree with the reviewer that: (a) One has to be cautious when using the dielectric constant to evaluate the Coulomb screening of excitons. (b) The exciton binding energy determined independently of the dielectric response is more accurate, e.g., from high field magnetoabsorption measurements. (c) The lattice screening of the exciton may be dynamic in nature.

We thank the reviewer for pointing out that in the introduction of our manuscript the discussion of exciton screening followed a misconception. As suggested by the reviewer, we included the following statements in the revised introduction: (a) The optical dielectric constant cannot describe the exciton binding energy in metal-halide perovskites.^[9] (b) Magnetoabsorption measurements^[1] are more accurate than the Elliott model in determining the exciton binding energy^[7]. (c). The lattice screening of the exciton is dynamic in nature^[10]. Furthermore, we included additional references (see below, indexed as in the MS):

[1]. Miyata, A. *et al.* Direct measurement of the exciton binding energy and effective masses for charge carriers in organic-inorganic tri-halide perovskites. *Nature Phys* **11**, 582–587; 10.1038/nphys3357 (2015).

[7]. Baranowski, M. & Plochocka, P. Excitons in Metal-Halide Perovskites. *Adv. Energy Mater.* **10**, 1903659; 10.1002/aenm.201903659 (2020).

[9]. Herz, L. M. How Lattice Dynamics Moderate the Electronic Properties of Metal-Halide Perovskites. *The journal of physical chemistry letters* **9**, 6853–6863; 10.1021/acs.jpclett.8b02811 (2018).

[10]. Miyata, K., Atallah, T. L. & Zhu, X.-Y. Lead halide perovskites: Crystal-liquid duality, phonon glass electron crystals, and large polaron formation. *Science advances* **3**, e1701469; 10.1126/sciadv.1701469 (2017).

The spectral resolution for the THz measurements in Fig. 3 looks heavily interpolated due to zero-padding the Fourier transform, which effectively Gaussian blurs each data point. This might wash over features in the spectrum due to the strong polar optical phonons that may be present in the system. The effect of this may be to assign phonon features to mobile carriers. The authors should replot their data without such over-padding to show the level of the noise and whether or not there are spectral features that might be due to optical phonons in the film. These phonon features have been seen in other THz studies on films.

Authors' response: We agree with the reviewer that zero-padding the Fourier transform (FT) can wash over optical phonon features. We applied the FT window (Hanning) instead of zero-padding, hence our data is free of interpolation effects however, the noise has been smoothed to some extent. FT windows are often applied to the time-domain waveforms to minimize the effect of frequency

leakage caused by the discontinuity at boundaries. For terahertz waveforms with pulse-shaped signals, the Hanning window is the best choice to resolve the frequency of interest by, (1) removing boundary discontinuity in the time domain, (2) smoothing the broadband random noise in the frequency domain (Figure R1):

Figure R1. (a) Photo-induced time-domain THz signal of CsFAMA thin film for a carrier density of $\sim 6.9 \times 10^{17} \text{ cm}^{-3}$. The time interval is 30 fs. The red dashed line indicates the FT window (Hanning). (b): FT magnitude of the original signal (blue open circles) and of the windowed signal (red solid line), respectively. The normalized FT magnitude in the frequency range from 0.4~2.3 THz is larger than 0.1. (c) Photo-induced change of photoconductivity (solid lines) and uncertainty (standard error, filled areas) determined from multiple measurements. The spectra between 0.4~2.3 THz exhibit less error, so this frequency range was selected for all THz spectra in this work. (d-f) The same spectra as (a-c) for a carrier density of $\sim 7.5 \times 10^{18} \text{ cm}^{-3}$.

The optical phonon effect on the photo-induced change of the photoconductivity spectra ($\Delta\sigma$) results mainly from the infrared-active TO-phonons:

a. Artifact due to data analysis:

$$\frac{\Delta E_{THz}}{E_{THz}} = -\frac{1}{\epsilon_0 c (1 + n_{sub})} \int_0^d \Delta \sigma(\omega, x) dx$$

The equation above was derived based on the zeroth-order approximation that the error occurs at frequencies with high dielectric background ^[S4], yielding artifacts in the calculated $\Delta\sigma$, especially at the peak of the TO-phonon oscillators. However, such artifact has only a small influence on the $\Delta\sigma$'s amplitude and thus the frequency dependence in metal-halide perovskites (Figure R2 ^[S4]).

Figure R2. Artifact in THz photoconductivity change due to zeroth-order approximation. Sample: MAPbI thin film.

[S4]. La-O-Vorakiat, C. *et al.* Phonon features in terahertz photoconductivity spectra due to data analysis artifact: A case study on organometallic halide perovskites. *Appl. Phys. Lett.* **110**, 123901; 10.1063/1.4978688 (2017).

b. Photo-induced change of TO-phonon resonances:

Photo-induced peak-shifts of the TO-phonon resonances were reported in CsPbBr nanocrystal thin film (Figure below, left ^[S5]), which has been ascribed to polaron effects. However, no obvious photo-induced peak shift and shape change of the TO-phonon resonances can be observed in our samples (Figure below, right, this work), except a small weakening of the resonance amplitude, which is usually caused by thermal expansion of the TO-phonons. This observation evidences that the contribution of TO-phonons to $\Delta\sigma$ is secondary:

Third-party Figure

See Fig. 2 in Ref:

Cinquanta, E. *et al.* Ultrafast THz Probe of Photoinduced Polarons in Lead-Halide Perovskites. *Phys. Rev. Lett.* **122**, 166601; 10.1103/PhysRevLett.122.166601 (2019).

Figure R3. Left: Peak shift of TO-phonon oscillators involved in large polaron formation in CsPbBr_3 [S5]. **Right:** The real part of the THz photoconductivity in the ground-state and excited state for CsFAMA, FAMA and MA thin films, respectively. Blue lines indicate the difference spectra ($\Delta\sigma$). No obvious peak shift and shape change of TO-phonon resonances are observed in our samples, except a small weakening of the resonance amplitude.

[S5]. Cinquanta, E. *et al.* Ultrafast THz Probe of Photoinduced Polarons in Lead-Halide Perovskites. *Phys. Rev. Lett.* **122**, 166601; 10.1103/PhysRevLett.122.166601 (2019).

We thank the reviewer for bringing to our attention that either FT methods or optical-phonons contribute to $\Delta\sigma$. As suggested by the reviewer, to demonstrate the level of our instrumental error, we added the standard error into the FT results shown in the revised manuscript (Fig. 3). To discuss the impact of optical-phonons on $\Delta\sigma$, we inserted a paragraph in the revised supplementary information. More precisely, we considered that the contribution of TO-phonons to our $\Delta\sigma$ spectra is minor. We included two more references to support this view in the corresponding revision

[S4,S5].

[S4]. La-O-Vorakiat, C. *et al.* Phonon features in terahertz photoconductivity spectra due to data analysis artifact: A case study on organometallic halide perovskites. *Appl. Phys. Lett.* **110**, 123901; 10.1063/1.4978688 (2017).

[S5]. Cinquanta, E. *et al.* Ultrafast THz Probe of Photoinduced Polarons in Lead-Halide Perovskites. *Phys. Rev. Lett.* **122**, 166601; 10.1103/PhysRevLett.122.166601 (2019).

Can the authors be a little more clear as to what the dipolar relaxation is the Debye model fitting? The text seems to suggest that it is related to the lattice, so is it a photoinduced relaxation of the cation orientation? It's hard to tell whether it's physically realistic or just that the model fits.

Authors' response: We agree with the reviewer that the discussion was not precise in regard to the contribution of the dipolar relaxation to the Debye relaxation. The success of the Debye model fits implies photo-induced dynamic disorder centered above ~ 4 THz. Since the cation dynamics are mainly active in the GHz region, the dipolar relaxation is related to the inorganic lattice. The difficulty here is, that the Debye relaxation requires infrared-active polar fluctuations, which are unlikely to exist in the region above ~ 4 THz^[75,76], indicating $\Delta\sigma$ is governed by other phenomena.

We thank the reviewer for bringing to our attention that the Debye relaxation may just be a parametrization. We now show that polar fluctuations are not responsible for the match to the Debye model^[75,76]. We have discussed other possible models in the revised manuscript, which are mathematically equivalent to the Debye model, but have different physical origin (see the revised manuscript). These models are carrier localization models related to lattice disorder, so they require lattice fluctuations only, regardless whether the lattice fluctuations are infrared-active or not. The most adequate model appears to be quantum Anderson localization^[24]. We included several references to support this view in our revision:

[24]. Lacroix, A., Laissardière, G. T. de, Quémerais, P., Julien, J.-P. & Mayou, D. Modeling of Electronic Mobilities in Halide Perovskites: Adiabatic Quantum Localization Scenario. *Phys. Rev. Lett.* **124**, 196601; 10.1103/PhysRevLett.124.196601 (2020).

[75]. Leguy, A. M. A. *et al.* Dynamic disorder, phonon lifetimes, and the assignment of modes to the vibrational spectra of methylammonium lead halide perovskites. *Physical chemistry chemical physics : PCCP* **18**, 27051–27066; 10.1039/C6CP03474H (2016).

[76]. Pérez-Osorio, M. A. *et al.* Vibrational Properties of the Organic–Inorganic Halide Perovskite CH₃NH₃PbI₃ from Theory and Experiment: Factor Group Analysis, First-Principles Calculations, and Low-Temperature Infrared Spectra. *J. Phys. Chem. C* **119**, 25703–25718; 10.1021/acs.jpcc.5b07432 (2015).

For the THz spectroscopy, how large was the pump beam spot at the sample? If the pump beam is too small compared to the frequency dependent THz spot size, there is a suppression in THz

conductivity at low frequencies that looks exactly like their data. Given that they are pumping with the output of a TOPAS and this output is typically only 1mm or less, this is a serious possibility. See for example Fig. 2 in the paper: M. C. Beard, G. M. Turner and C. A. Schmuttenmaer, Phys. Rev. B 62, 15764 (2000) where they show a strong suppression for a pump spot of 1.1 mm diameter. The authors should confirm that their data shows no such dependence on the pump spot size and it is truly a homogeneous excitation for all frequencies in the THz pulse bandwidth. Did the authors check that a Drude model was observed in the same experimental conduction using a typical Drude semiconductor like GaAs, for instance?

Authors' response: The reviewer is right that the pump beam spot can significantly affect the frequency-dependent $\Delta\sigma$. In our terahertz setup, the spot size of the terahertz probe at the focal point was estimated to ~ 2.4 mm in D86, or 1.4 mm in FWHM. To avoid any spot-size effect and to maximize the photon fluence, the pump diameter was set to ~ 3.992 mm (D86), as measured with a beam profiler:

Figure R4. The diameter of our THz experiment pump beam measured by a beam profiler.

Below we shows the $\Delta\sigma$ of an n-doped silicon wafer (resistivity $\sim 1-3 \Omega$, thickness $\sim 283 \mu\text{m}$) for different excitation fluence. As expected, the datasets can all be fitted by the Drude model (Figure below), indicating our terahertz setup is free of any spot-size effect.

Figure R5. $\Delta\sigma$ of a n-doped silicon wafer (resistivity $\sim 1\text{-}3\ \Omega$, thickness $\sim 283\ \mu\text{m}$) at different pump fluence. Solid lines show Drude fits to the data, indicating that our THz experiment is free of pump size effects. The fit results are shown in the table.

As suggested by the reviewer, we inserted a paragraph discussing the spot-size effect in the supplementary information. Furthermore, we included the reference below to support the corresponding revisions [S3];

[S3] Beard, M. C., Turner, G. M. & Schmuttenmaer, C. A. Transient photoconductivity in GaAs as measured by time-resolved terahertz spectroscopy. *Phys. Rev. B* **62**, 15764–15777; 10.1103/PhysRevB.62.15764 (2000).

The data should have some indication of measurement uncertainty. This could easily be done by presenting the mean and the standard error of the mean obtained from analysis of the multiple time-domain waveforms. The fits should then take into account the measurement uncertainty,

which will be greater at the lowest and highest of their frequency window. There is quite a bit of smoothing in the data.

Authors' response: We agree with the reviewer that the measurement uncertainty is greater at the low and high ends of the frequency window, and that the fits should indicate the measurement uncertainty.

As suggested by the reviewer, we added to the revised manuscript the standard errors of the FT results of the THz signals. The fits have been weighted by these instrumental errors.

The authors make a connection between the change in the Debye contribution and the electron-phonon coupling strength using a polaron mobility model, taking into account the expected increase in the effective mass and therefore mobility upon dressing with LO phonons. They find the fitted static dielectric permittivity is less than the theoretically predicted value, however has this not been measured? I'm also confused as to why they say that optical phonons contribute weakly to the dielectric response. These are highly polar materials and the optical phonons contribute significantly to the dielectric response. Perhaps they meant the LO phonons? Even then the Lyddane-Sachs-Teller relation connects the LO phonon frequency to the static permittivity, so one can estimate the contributions.

Authors' response: The reviewer is right that experimental data of the static dielectric permittivity is available, and that the optical phonons contribute significantly to the dielectric response. Hence, our previous statements about the contribution of optical phonons to the static dielectric response was inappropriate. The dielectric response accounting for all optical phonons is ~16.5-28.5 as reported in the literature (Figure R6, ϵ_{ion})^[3].

[3] Wilson, J. N., Frost, J. M., Wallace, S. K. & Walsh, A. Dielectric and ferroic properties of metal halide perovskites. *APL Materials* **7**, 10901; 10.1063/1.5079633 (2019).

Third-party Figure

See Tab. 1 in Ref:

Wilson, J. N., Frost, J. M., Wallace, S. K. & Walsh, A.
Dielectric and ferroic properties of metal halide perovskites.
APL Materials **7**, 10901; 10.1063/1.5079633 (2019).

Figure R6. Dielectric response of halide perovskites.

As suggested by the reviewer, we removed the inappropriate statements in the revised manuscript. In fact, we removed the discussion about the polaron mobility, because we noticed that: (a) there is an ongoing debate about the polaron picture at room T ^[24]; (b) nonlinear-Fröhlich coupling plays an important role in the carrier transport^[26]. We included several references to support the corresponding revisions:

[24]. Lacroix, A., Laissardière, G. T. de, Quémerais, P., Julien, J.-P. & Mayou, D. Modeling of Electronic Mobilities in Halide Perovskites: Adiabatic Quantum Localization Scenario. *Phys. Rev. Lett.* **124**, 196601; 10.1103/PhysRevLett.124.196601 (2020).

[26]. Mayers, M. Z., Tan, L. Z., Egger, D. A., Rappe, A. M. & Reichman, D. R. How Lattice and Charge Fluctuations Control Carrier Dynamics in Halide Perovskites. *Nano letters* **18**, 8041–8046; 10.1021/acs.nanolett.8b04276 (2018).

More on the editorial side, there appears to be something wrong with the bibliography. The parameters related to the polaron formation seem to be taken from Ref. 39, however no reference exists in the text. Instead the authors refer to Ref 36 on line 203 which concerns exciton-Mott physics unrelated to perovskites. Reference 40 is referred to when discussing polaron formation (line 205) however I believe this should be Refs 39 that measures the polaron formation in MAPI and 41 which investigates the frequency averaged risetime with excess energy and assigns it to polaron formation (mentioned in line 211).

Authors' response: We thank the reviewer for bringing to our attention the editorial mistakes in the previous version of the manuscript. We revised the manuscript accordingly.

Reviewer #2 (Remarks to the Author):

In their manuscript, Wang et al present an experimental study on the ultrafast relaxation of lead halide perovskite. Using a combination of transient absorption and time domain terahertz spectroscopy, they study the band-gap narrowing following an optical excitation, and claim that their observations are uniquely explained by a photo-induced, transient increase in electron-phonon coupling. While the authors results are interesting, I find the analysis not entirely convincing. Mostly, this is due to implications that seem inconsistent with other observations in the literature and an insufficient discussion of the underlying mechanism. As such, I believe publication at this time would be premature.

Authors' response: We thank the reviewer for the time and effort spent to review our work, and for the constructive comments, which we address point-by-point below.

The authors might consider the following as they revise their manuscript:

There is very little discussion of temperature induced broadening in the manuscript and no attempt to rule it out as the reason for the observed band gap narrowing. First, the static broadening contribution is not discussed sufficiently. The SI states the functional form used, but not the parameters nor a study their sensitivity. Is σ changed in the excited state? Further, the excitation is significantly above the band gap and thus there is a significant amount of energy that must be dissipated by the lattice. If polarons are formed, there must be a large energy transfer to the lattice. However, perovskites are notoriously poor thermal conductors. Can local heating explain the band gap renormalization (either directly, or indirectly due to local lattice expansion and accompanying strain)?

Authors' response: We agree with the reviewer that the discussion on thermal effects in our previous manuscript is not sufficient. Below we try to clarify the presence of thermal effects.

Is sigma changed in the excited state?

Sigma possibly changes with carrier density and carrier temperature (Figure R7).

Figure R7. Absorption coefficient after photoexcitation for (a) $t=5 \text{ ps}$ for two different carrier densities. (b) $N \sim 3.5 \times 10^{18} \text{ cm}^{-3}$, at different time delays (different carrier temperature). The bottom panels of (a) and (b) show the photo-induced changes.

The extraction of the change in sigma needs fits to the transient absorption spectra. Phenomenologically, sigma is determined from the band edge absorption, which is also affected by exciton screening and the Fermi-Dirac distribution of hot carriers, resulting in many fit parameters (>6) and thus the extracted sigma needs to be carefully evaluated. Our approach here to extract the bandgap renormalization is to analyze the high-energy part of the transient absorption spectra neglecting changes in sigma. Our equation yields large errors near the band edge (Figure

R8) however, only small errors ($\sim 1\%$) around $E=1.82$ eV, which corresponds to the highest energy where band dispersion can still be described by a parabola. Hence, we believe it is safe to neglect photo-induced changes of sigma, when discussing the high-energy part of the TA signals.

Figure R8. Deviation of our simplified equation to describe changes of the photo-induced absorption. Parameters are: $E_g=1.6769$ eV, $R_{b0}=11.3$ meV, $\Delta R_b=1$ meV, $\Delta E_{bgf}=-5$ meV, $T_e=350$ K, $E_F=1.62$ eV. Inset: the ‘exact’ curve was calculated by using a hyperbolic secant broadening function with an initial width of 50 meV, broadened to 60 meV, while the approximated curve was calculated by our simplified equation, which is independent of the broadening function.

Can local heating explain the band gap renormalization (either directly, or indirectly due to local lattice expansion and accompanying strain)?

The thermal effect on bandgap renormalization is negligible in our experiments. First, heat accumulation caused by multi-pulse excitation can be ruled out, because the pulse interval ($\sim 333 \mu\text{s}$) is much longer than the carrier lifetime (< 200 ps in our samples). Second, the temperature increase caused by photoexcitation is smaller than 0.32 K for a carrier density of $3.5 \times 10^{18} \text{ cm}^{-3}$

photogenerated by 550 nm photons (~2.25 eV), as calculated from the heat capacity of the MHP lattice (~170-190 J/K/mol at room temperature).^[65] We used the reference below indexed here as in the revised manuscript to calculate the increase in T:

[65] Onoda-Yamamuro, N., Matsuo, T. & Suga, H. Calorimetric and IR spectroscopic studies of phase transitions in methylammonium trihalogenoplumbates (II)†. *Journal of Physics and Chemistry of Solids* **51**, 1383–1395; 10.1016/0022-3697(90)90021-7 (1990).

$$\Delta T_l = \frac{\Delta E_{ex} NV}{C_{MHP} * Mol} = \frac{(2.25 \text{ eV} - 1.6 \text{ eV}) * 1.6e19 \text{ J/eV} * 3.5e18 \text{ cm}^{-3} * 1 \text{ cm}^3}{170 \frac{\text{J}}{\text{K} \cdot \text{mol}} \cdot \frac{4e21}{N_A} \text{ mol}} \cong 0.32 \text{ K}$$

where ΔE_{ex} is the excess energy of pump photons, N is the carrier density, V is the volume in cm^3 . C_{MHP} is the molar heat capacity of the MHP lattice, where we used the lower-bound value of 170 J/K/mol. 4×10^{21} is the number of MHP unit cells in $V=1 \text{ cm}^3$. N_A is the Avogadro constant.

Lastly, the overall effect of the temperature increase results in a linear bandgap broadening instead of bandgap narrowing in perovskites (Figure R9, ~0.3 meV/K^[4]), yielding a bandgap increase smaller than ~0.1 meV in our experiments. Therefore, thermal effects cannot explain the linear bandgap narrowing observed in our work.

Figure R9. Energy gap of MAPbI₃ determined from T-dependent absorption spectra via Elliott model. E_g increases with temperature, which was ascribed to the inverse ordering of conduction bands.

[4]. Davies, C. L. *et al.* Bimolecular recombination in methylammonium lead triiodide perovskite is an inverse absorption process. *Nature communications* **9**, 293; 10.1038/s41467-017-02670-2 (2018).

As suggested by the reviewer, we added the discussion of possible thermal effects to the revised manuscript. More precisely, we evidenced that the analysis of photo-induced absorption at $E=1.82$ eV can safely neglect the broadening function. Furthermore, the photo-induced temperature increase of the MHP lattice was estimated to ~ 0.32 K for the highest carrier density ($\sim 3.5 \times 10^{18}$ cm⁻³), resulting in a bandgap increase of ~ 0.1 meV, which cannot explain the linear bandgap narrowing observed by us. We included two more references to support the corresponding revisions ^[4,65]:

[4]. Davies, C. L. *et al.* Bimolecular recombination in methylammonium lead triiodide perovskite is an inverse absorption process. *Nature communications* **9**, 293; 10.1038/s41467-017-02670-2 (2018).

[65] Onoda-Yamamuro, N., Matsuo, T. & Suga, H. Calorimetric and IR spectroscopic studies of phase transitions in methylammonium trihalogenoplumbates (II)†. *Journal of Physics and Chemistry of Solids* **51**, 1383–1395; 10.1016/0022-3697(90)90021-7 (1990).

Many estimates of polaron binding energies are $\sim kT$ [1,2], which are consistent with the present authors study (line 204). However such a weak binding would not result in a significant population of localized carriers. Indeed, many other estimates of the number of free carriers report it to be much higher under these excitation densities [3]. How do the authors justify this discrepancy, both conceptually with the low binding energy and experimentally with the inconsistency with previous reports.

1. Bischak, Connor G., et al. "Tunable polaron distortions control the extent of halide demixing in lead halide perovskites." *The journal of physical chemistry letters* 9.14 (2018): 3998-4005.
2. Frost, Jarvist Moore. "Calculating polaron mobility in halide perovskites." *Physical Review B* 96.19 (2017): 195202.
3. Wehrenfennig, et al. *Advanced materials* (2014)

Authors' response: We agree with the reviewer that large polarons cannot result in a significant population of localized carriers, and free carriers are the main species photogenerated at room temperature. The significant carrier localization ($\sim 90\%$) observed by us is not precise. The value of $\sim 90\%$ originates from the large uncertainty of the Drude scattering time due to the small frequency-dependence of the THz photoconductivity. In the revised manuscript, we addressed this issue by fixing the carrier density to the measured value.

We removed the corresponding statements in the revised manuscript. Furthermore, new fits have been performed based on the assumption that all photo-generated carriers are free carriers. We included several more references to support the corresponding revisions [22,49,86].

[22] Frost, Jarvist Moore. "Calculating polaron mobility in halide perovskites." *Physical Review B* 96.19 (2017): 195202.

[49] Wehrenfennig, et al. *Advanced materials* (2014)

[86] Bischak, Connor G., et al. "Tunable polaron distortions control the extent of halide demixing in lead halide perovskites." *The journal of physical chemistry letters* 9.14 (2018): 3998-4005.

Finally, there is no attempt to associate a mechanism to the increased electron-phonon coupling. Typically, the static dielectric is decreased with increased carriers as polarization fluctuations (which determine the dielectric constant) become constrained. To have a photo-induced effect that was not just from heating, the authors would have to posit the emergence of a novel motion enabled under excitation. Further, the authors discussion is limited to a simple model of electron phonon coupling. A significant amount of work has gone into the study of this coupling, demonstrating that it is not well described by a Frohlich model [4,5] The authors would benefit from considering, and ruling out, contributions not contained in this model.

4. Mayers, Matthew Z., et al. "How lattice and charge fluctuations control carrier dynamics in halide perovskites." *Nano letters* 18.12 (2018): 8041-8046.

5. Schlipf, Martin, Samuel Poncé, and Feliciano Giustino. "Carrier lifetimes and polaronic mass enhancement in the hybrid halide perovskite CH₃NH₃PbI₃ from multiphonon Fröhlich coupling." *Physical review letters* 121.8 (2018): 086402.

Authors' response: We agree with the reviewer that the discussion of the underlying mechanism of increased electron-phonon coupling was lacking in our previous manuscript. We noticed that a photo-induced increase of the dielectric constant is unusual. Our THz data revealed lattice fluctuations centered above ~4 THz however, no corresponding polar fluctuations could be observed, indicating another mechanism. Here, we benefit a lot from the reviewer's excellent suggestion. Indeed, other models, mathematically equivalent to the Debye model, can also fit our THz photoconductivity. However, these models rely on different physics (see the revised manuscript). The most applicable model is the quantum Anderson localization^[24], which only

requires lattice fluctuations, regardless of whether the lattice fluctuations are infrared-active or not. While our techniques cannot directly detect lattice fluctuations, evidence of photo-induced lattice fluctuations has been reported in other works in the mid-IR^[30] as well as in works using ultrafast electron diffraction^[50].

As suggested by the reviewer, we revised the discussion about the THz photoconductivity. More precisely, we ruled out the explanation of a photo-induced dielectric response. Instead, we considered the possibility of quantum Anderson localization, which only requires lattice fluctuations, regardless of whether they are infrared-active or not. The presence of photo-induced lattice fluctuations is supported by earlier work in the mid-IR^[30] and ultrafast electron diffraction^[50]. Since the lattice fluctuations contribute to the nonlinear-Fröhlich coupling^[26], we removed the discussion of polaron mobility based on the conventional Fröhlich coupling in the revised manuscript. We included several references to support the corresponding revisions:

[13]. Schlipf, M., Poncé, S. & Giustino, F. Carrier Lifetimes and Polaronic Mass Enhancement in the Hybrid Halide Perovskite $\text{CH}_3\text{NH}_3\text{PbI}_3$ from Multiphonon Fröhlich Coupling. *Phys. Rev. Lett.* **121**, 86402; 10.1103/PhysRevLett.121.086402 (2018).

[24]. Lacroix, A., Laissardière, G. T. de, Quémerais, P., Julien, J.-P. & Mayou, D. Modeling of Electronic Mobilities in Halide Perovskites: Adiabatic Quantum Localization Scenario. *Phys. Rev. Lett.* **124**, 196601; 10.1103/PhysRevLett.124.196601 (2020).

[26]. Mayers, M. Z., Tan, L. Z., Egger, D. A., Rappe, A. M. & Reichman, D. R. How Lattice and Charge Fluctuations Control Carrier Dynamics in Halide Perovskites. *Nano letters* **18**, 8041–8046; 10.1021/acs.nanolett.8b04276 (2018).

[30]. Munson, K. T., Kennehan, E. R., Doucette, G. S. & Asbury, J. B. Dynamic Disorder Dominates Delocalization, Transport, and Recombination in Halide Perovskites. *Chem* **4**, 2826–2843; 10.1016/j.chempr.2018.09.001 (2018).

[50]. Wu, X. *et al.* Light-induced picosecond rotational disordering of the inorganic sublattice in hybrid perovskites. *Science advances* **3**, e1602388; 10.1126/sciadv.1602388 (2017).

Reviewer #3 (Remarks to the Author):

In this paper the authors report photo-induced enhancement of electron-phonon coupling and dynamic disorder in triple cation mixed halide perovskite using ultrafast transient absorption and THz spectroscopies. The authors' conclusions are based on observation of linear dependence of photoinduced band-gap normalization on the carrier density and Debye relaxation component observed in the terahertz photoconductivity spectra. This paper reports results of great general interest. Unfortunately, it overlooks several results that are not explained and commented upon. The authors present the portion of the data and fittings. I would find it clearer for the authors to first present all the data and fitting curves (with error estimations) so that the reader gets a clear view of the interpretation and analysis. My concerns and suggestions are detailed below.

Authors' response: We thank the reviewer for the time and effort spent to review our work, and for the constructive comments, which we try to address point-by-point below.

1) While describing the high-energy tails in transient absorption spectra, authors have mentioned that “ $R_b=R_{b0}$ was used in the fitting, since even a small reduction in Rb resulted in poor fits” (Page 7, line 144). This fitting has a huge importance to the authors' conclusions as it directly relates to the red-shifted excitonic level. So it is essential that the authors mention the parameter space tried for fitting along with the error estimations.

Authors' response: We agree with the reviewer that the exciton screening is essential to the conclusion drawn from the fits to the TA spectra. We set $R_b=R_{b0}$ in the fit because a small reduction in Rb resulted in poor fits of the entire TA spectra (~1.5-1.9 eV). We added two references to support this approach (see also Figure R10) ^[57,58] Reference indexes are the indexes as in the revised manuscript:

Figure R10. Left: Fits to the TA spectra using constant exciton binding energy ^[57]. Right: Another example: fits to the TA spectra using a constant exciton binding energy ^[58].

[57] Lim, J. W. M. *et al.* Hot Carriers in Halide Perovskites: How Hot Truly? *The journal of physical chemistry letters* **11**, 2743–2750; 10.1021/acs.jpcclett.0c00504 (2020).

[58] Price, M. B. *et al.* Hot-carrier cooling and photoinduced refractive index changes in organic-inorganic lead halide perovskites. *Nature communications* **6**, 8420; 10.1038/ncomms9420 (2015).

Since the fits to the entire TA spectra require many free parameters (>6), the fit has many points where it converges, which requires that the results are carefully evaluated. Our approach to determine the bandgap renormalization is to analyze the high-energy part of the TA spectra. Our equation yields large errors near the band edge (Figure R8) however, only small errors (~1%) around $E=1.82$ eV, which is the highest energy at which the band dispersion can still be described by a parabola. (see Figure R11).

Figure R11. Deviation of our simplified equation to describe photo-induced changes of absorption. Parameters are: $E_g=1.6769$ eV, $R_{b0}=11.3$ meV, $\Delta R_b=1$ meV, $\Delta E_{bgr}=-5$ meV, $T_e=350$ K, $E_F=1.62$ eV. Inset: the ‘exact’ curve was calculated using a hyperbolic secant broadening function with an initial width of 50 meV broadened to 60 meV, while the ‘approximated’ curve was calculated by our simplified equation, which is independent on the broadening function.

Indeed, the fits to the high-energy part of the TA spectra are affected by the exciton screening. To clarify the impact of exciton screening, we further simplified our equation by Taylor expansion neglecting the higher order terms (see SI for the detailed derivation):

$$\Delta\alpha(E) \cong \frac{A}{E} \xi_0 \sqrt{x} \left[\left(\xi_0 e^{-2\pi\sqrt{\frac{R_{b0}}{x}}} - 1 \right) \frac{\Delta R_b}{2R_{b0}} - \xi_0 e^{-2\pi\sqrt{\frac{R_{b0}}{x}}} \frac{\Delta E_{bgr}}{2x} \right]$$

where $\xi_0 e^{-2\pi\sqrt{R_{b0}/x}} \leq 1$ with a constant ξ_0 . Because the exciton screening (ΔR_b) is non-negative, ΔE_{bgr} should be negative for photo-induced absorption around 1.82 eV ($\Delta\alpha > 0$). Since a blue-shift of the photo-bleach peak due to band filling is observed in the TA spectra (Figure R12), the absolute value of ΔE_{bgr} should be smaller than the Burstein-Moss shift, which can be estimated from the width of the broadened photo-bleach peak (~ 38 meV for $N=3.5 \times 10^{18}$ cm $^{-3}$). Using $\Delta E_{bgr} = -38$ meV, the upper limit of ΔR_b calculated from the equation above is $\sim 0.1R_{b0}$, indicating a red-shift of the excitonic level. We hypothesize that the small ΔR_b is related to the formation of

Mahan excitons ^[60], whose binding energy increases almost linearly with carrier density ^[61]. In this case, the linear ΔE_{bgr} dependence mentioned in our previous manuscript remains valid.

[60] Palmieri, T. *et al.* Mahan excitons in room-temperature methylammonium lead bromide perovskites. *Nature communications* **11**, 850; 10.1038/s41467-020-14683-5 (2020).

[61] Schleife, A., Rödl, C., Fuchs, F., Hannewald, K. & Bechstedt, F. Optical absorption in degenerately doped semiconductors: Mott transition or Mahan excitons? *Phys. Rev. Lett.* **107**, 236405; 10.1103/PhysRevLett.107.236405 (2011).

Figure R12. The peak shift and broadening of the photo-bleach due to band filling.

We removed the assumption of ‘Rb=Rb0’ and the fits of the high energy tails in the revised manuscript, because the exciton screening contains a big uncertainty. Instead, we applied a different strategy and reached the same conclusion, that is, the absolute value of ΔE_{bgr} increases linearly with carrier density, as stated before. We included several references to support the corresponding revisions:

[57] Lim, J. W. M. *et al.* Hot Carriers in Halide Perovskites: How Hot Truly? *The journal of physical chemistry letters* **11**, 2743–2750; 10.1021/acs.jpcllett.0c00504 (2020).

[58] Price, M. B. *et al.* Hot-carrier cooling and photoinduced refractive index changes in organic-inorganic lead halide perovskites. *Nature communications* **6**, 8420; 10.1038/ncomms9420 (2015).

[60] Palmieri, T. *et al.* Mahan excitons in room-temperature methylammonium lead bromide perovskites. *Nature communications* **11**, 850; 10.1038/s41467-020-14683-5 (2020).

[61] Schleife, A., Rödl, C., Fuchs, F., Hannewald, K. & Bechstedt, F. Optical absorption in degenerately doped semiconductors: Mott transition or Mahan excitons? *Phys. Rev. Lett.* **107**, 236405; 10.1103/PhysRevLett.107.236405 (2011).

2) Authors have mentioned that “For $N \sim 4.3 \times 10^{18} \text{ cm}^{-3}$, ΔE_{bgr} assumes a plateau between ~ 2 - 5 ps after reaching the maximum (Fig. 2a), indicating photo-induced BGR is concluded in ~ 2 ps” (Page 9, line 170). With all due respect, I don’t see that in Fig. 2a. ΔE_{bgr} seems to maximize at ~ 700 - 800 fs and then shows a decay followed by ill-defined dynamics in 2 - 5 ps region, which is defined as plateau by the authors. Similar plateau is also observed in 10 - 30 ps region. Can authors clarify this? Also, what is the cause of decay of ΔE_{bgr} after reaching the maximum? Can it be contributed by carrier-carrier interactions?

Authors’ response: We agree with the reviewer that the discussion of the time-dependent ΔE_{bgr} in the previous manuscript was questionable. The fitted $\Delta E_{bgr}(t)$ shown in our previous manuscript was obtained with ‘ $R_b = R_{b0}$ ’. As mentioned above (our response to the reviewer’s point 1), the assumption of ‘ $R_b = R_{b0}$ ’ is inappropriate, because exciton screening does exist. Unfortunately, ΔR_b and ΔE_{bgr} cannot be disentangled, because they are mutually dependent (positively correlated) in the fit, so we changed our strategy to extract the N -dependence of ΔE_{bgr} in the revised manuscript.

Next, we address the reviewer’s question about the plateau and the sub-ps decay of ΔE_{bgr} based on the assumption of ‘ $R_b = R_{b0}$ ’, though these issues do no longer exist in the revised manuscript.

Figure R13. The fitted $\Delta E_{bgr}(t)$ based on $R_b=R_{b0}'$. **Left:** The fitted $\Delta E_{bgr}(t)$ of CsFAMA thin films from the previous version of the manuscript. **Right:** The fitted $\Delta E_{bgr}(t)$ for different photon energies, carrier densities, and samples.

We agree with the reviewer that the plateau from 2-5 ps is not obvious (Figure R13, left). Recently, we systematically studied $\Delta E_{bgr}(t)$ by using different photon energies, carrier density, and perovskite samples (Figure R12, right). All $\Delta E_{bgr}(t)$ are featureless and their normalized $\Delta E_{bgr}(t)$ are virtually the same. The discrepancy between the two experiments can originate from sample differences, leading to different higher order recombination rates.

We thank the reviewer for bringing to our attention that the discussion of the time-dependent ΔE_{bgr} in the manuscript was questionable. We have removed this discussion in the revised manuscript.

3) It is important that the authors show the time-dependent BGR for other carrier densities in supporting information. It might also help the authors to answer questions raised by me in point 2.

Authors' response: We agree with the reviewer that the dependence of $\Delta E_{bgr}(t)$ on carrier density can strongly support our previous conclusion. Please see our response to the reviewer's question in point 2.

4) Authors have also performed transient absorption measurements on FAMARb and FAMA. Again, it will be useful to present time-dependent BGR data for these systems as well. Does the $Rb=Rb_0$ fitting also holds for these systems (point 1). Please show the fitted data in the supporting information. It is very unfortunate that the data on these two systems have not been given their due importance. In fact, it has only been mentioned once in the main manuscript. Also, authors have not provided the rationale behind probing these specific systems.

Authors' response: We agree with the reviewer that determining $\Delta E_{bgr}(t)$ on different samples can strongly support our conclusion. The high energy tail fit to the conventional Boltzmann equation requires only 1 free parameter (T_e), thus it is easier to use than our model which requires 3 free parameters (E_F , T_e and ΔE_{bgr}), regardless of the perovskite system. The fitted $\Delta E_{bgr}(t)$ for different perovskite systems is shown in Fig. R13. Actually, 'Rb=Rb0' holds not only for the high-energy tail fitting, but also for the full spectra fitting (Fig. R10, Authors' response to the reviewer's point 1). The success of the 'Rb=Rb0' hypothesis could be related to the formation of Mahan excitons.

The reason why we selected CsFAMA, FAMA, and MAPbI systems is that they are prominent candidates for applications in single-junction perovskite solar cells, with similar optical bandgap $E_{opt} \sim 1.6$ eV. We tried to reveal the impact of different cations on the optoelectronic properties. However, a pronounced difference in the carrier recombination was not observed in our earlier work^[56]. Hence, we studied the N -dependence of the bandgap renormalization and carrier mobility.

[56]. Gao, Y. *et al.* Impact of Cesium/Rubidium Incorporation on the Photophysics of Multiple-Cation Lead Halide Perovskites. *Sol. RRL* **4**, 2000072; 10.1002/solr.202000072 (2020).

We thank the reviewer for bringing to our attention that $\Delta E_{bgr}(t)$ of different samples can strongly support our conclusion. The approach to determine ΔE_{bgr} has changed in the revised manuscript, thus the fits to the high-energy tails of the TA spectra resulting in questionable $\Delta E_{bgr}(t)$ are no longer part of the revised manuscript. However, we present the entire transient absorption spectra (1.5-2 eV) for all perovskite systems in the revised supporting information.

5) Authors have used different pump wavelengths for transient absorption (2.61 eV) and tdTHz (2.25 eV). Can authors please explain the reason for this? It might result into different dynamics and coupling to different phonons at different excitation frequencies.

Authors' response: We agree with the reviewer that different pump wavelengths can result in different carrier dynamics. We measured transient absorption spectra (2.61 eV) on the samples prior to measuring the THz spectra. The reason why we changed to 2.25 eV excitation for the THz measurements is that this photon energy was best to obtain THz signals (because of beam quality and fluence). Indeed, the pump wavelength has a strong influence on the sub-ps dynamics both in transient absorption ^[58] and THz experiments ^[68], but only a small influence on the carrier dynamics after hot-carrier cooling ^[58,68]. We had not measured the transient absorption with 2.25 eV photons, because our study does not focus on the sub-ps dynamics. To address this issue, we conducted an additional experiment using 2.25 eV pump photons in both transient absorption and THz spectroscopy. We investigated the influence of the pump photon energy on the sub-ps dynamics by analyzing the transient absorption spectra and included further references to the revised manuscript.

[58] Price, M. B. *et al.* Hot-carrier cooling and photoinduced refractive index changes in organic-inorganic lead halide perovskites. *Nature communications* **6**, 8420; 10.1038/ncomms9420 (2015).

[68] Bretschneider, S. A. *et al.* Quantifying Polaron Formation and Charge Carrier Cooling in Lead-Iodide Perovskites. *Advanced materials (Deerfield Beach, Fla.)*, e1707312; 10.1002/adma.201707312 (2018).

A discussion of the transient absorption and THz spectra when using the same pump photon energy (2.25 eV) has been included in the revised manuscript. We have also inserted a paragraph to discuss the influence of the carrier's excess energy to the revised manuscript. Furthermore, we included several references to support the corresponding revisions:

[58] Price, M. B. *et al.* Hot-carrier cooling and photoinduced refractive index changes in organic-inorganic lead halide perovskites. *Nature communications* **6**, 8420; 10.1038/ncomms9420 (2015).

[68] Bretschneider, S. A. *et al.* Quantifying Polaron Formation and Charge Carrier Cooling in Lead-Iodide Perovskites. *Advanced materials (Deerfield Beach, Fla.)*, e1707312; 10.1002/adma.201707312 (2018).

6) Can authors provide tdTHz data on other two perovskite systems as well? This will really strengthen the inferences drawn using combination of the classical Drude model and the Debye relaxation model to describe fluence-dependent THz photoconductivity spectra.

Authors' response: We agree with the reviewer that data from different perovskite systems can further support our conclusion. As suggested by the reviewer, we added relevant data and fits to the supporting information. Lastly, we made an extensive revision of the fit model used in the revised manuscript, since the Debye model is not self-consistent (see the revised manuscript).

Minor points:

7) Main manuscript and Supplementary information have different titles.

Authors' response: Revised.

8) Page 6, line 123: carrier density should be 3.6×10^{16} and not 3.6×10^{18} .

Authors' response: Revised.

9) Page 2, line37: excitons are observed at room temperature. Refs: Nature Communications volume 11, Article number: 850 (2020); ACS Photonics 2018, 5, 3, 852–860.

Authors' response: We agree with the reviewer that excitons have been observed at room temperature. Our previous hypothesis that 'Rb=Rb0' is in good agreement with the formation of Mahan excitons. As suggested by the reviewer, we applied the concept of Mahan excitons to explain the photo-induced red-shift of the excitonic level in the revised manuscript. We cited both references in the revised manuscript.

[16]. Jha, A. *et al.* Direct Observation of Ultrafast Exciton Dissociation in Lead Iodide Perovskite by 2D Electronic Spectroscopy. *ACS Photonics* **5**, 852–860; 10.1021/acsp Photonics.7b01025 (2018).

[60]. Palmieri, T. *et al.* Mahan excitons in room-temperature methylammonium lead bromide perovskites. *Nature communications* **11**, 850; 10.1038/s41467-020-14683-5 (2020).

10) Figure 3a inset is hard to comprehend.

Authors' response: We moved the time-domain THz spectra to the supporting information of the revised manuscript. More details of the time-domain THz spectra are now given in the revised supporting information.

11) Authors might also consider to refer following papers at appropriate places in the introduction: Science Advances 2019, Vol. 5, no. 5, eaaw5558; Journal of Applied Physics 124, 215106 (2018).

Authors' response: The additional references have been cited.

[44]. Lan, Y. *et al.* Ultrafast correlated charge and lattice motion in a hybrid metal halide perovskite. *Science advances* **5**, eaaw5558; 10.1126/sciadv.aaw5558 (2019).

[45]. Kumar, A. *et al.* Ultrafast THz photophysics of solvent engineered triple-cation halide perovskites. *Journal of Applied Physics* **124**, 215106; 10.1063/1.5051561 (2018).

REVIEWER COMMENTS

Reviewer #1 (Remarks to the Author):

The authors have addressed my concerns in this new manuscript.

One thing to note in the response to the second reviewer's comments, the authors state heating is not likely as the pump repetition period is much longer than the 200 ps carrier lifetime. However, it is the phonon lifetime that matters in this case and ultimately the transport of the energy out of the sample, limited by the thermal conductivity of the sample and substrate. They calculate a temperature change due to one pulse as small, however accumulated heating over many pulses may still be a problem.

If the authors can convincingly address this question, I can support publication.

Reviewer #2 (Remarks to the Author):

The authors have done an admirable job of revising their manuscript based on the comments and critiques of the reviewers. Notably they have nearly completely revised their interpretation of the bandgap narrowing and transient frequency dependent photocurrent. However, the provided explanations invoking Anderson localization + hopping, and Mahan excitons seem similarly ad hoc as their previous explanations, and inconsistent with the larger body of work of the lead halides. For example, if carriers were localized and transport was dominated by hopping, the variable range hopping model of Mott predicts a temperature dependence inconsistent with that observed experimentally (exponential rather than the observed power law). The $T^{-3/2}$ is the naive expectation from acoustic phonon scattering, so why this is not discussed or ruled out? Mahan excitons have only recently and found to be very sensitive to the exciton binding energy (observed in MAPbBr₃ but expected in MAPbI₃ Ref 60) so if the authors are claiming their importance here, they ought to have additional evidence beyond correlative speculation. If such extrapolations beyond what their data is able to furnish are tamed, I would be in favor of publication.

Reviewer #3 (Remarks to the Author):

Wang et al have diligently addressed the numerous scientific and technical issues raised in my previous review, and I now find this work to be highly convincing and comprehensive and publishable in Nature Communications. Congratulations on some very nice work!

While I do not require further review of the manuscript, I would strongly recommend that the authors discuss the comparison of photophysics in triple-cation MHPs (studied in the current manuscript) with other reported conventional MHPs to draw broader conclusions. Also, the referencing style is not consistent, which should be corrected.

Dr. Ajay Jha

Reviewer #1 (Remarks to the Author):

The authors have addressed my concerns in this new manuscript. One thing to note in the response to the second reviewer's comments, the authors state heating is not likely as the pump repetition period is much longer than the 200 ps carrier lifetime. However, it is the phonon lifetime that matters in this case and ultimately the transport of the energy out of the sample, limited by the thermal conductivity of the sample and substrate. They calculate a temperature change due to one pulse as small, however accumulated heating over many pulses may still be a problem. If the authors can convincingly address this question, I can support publication.

Authors' response: We thank the reviewer for his/her encouraging comments on our work. The '200 ps' carrier lifetime was a typo, it was supposed to be '200 ns'. The reviewer correctly states that heat accumulation cannot be entirely ruled out by simply comparing the carrier lifetime with the off time between laser pulses, in other words, the pump pulse repetition rate. In the pertinent literature, the heat transport time to the substrate (fused silica, which is less ideal than the quartz substrate we used in our experiments) for perovskite films has been reported to be $<9 \mu\text{s}$ (Ščajev, P. *et al.* Anisotropy of Thermal Diffusivity in Lead Halide Perovskite Layers Revealed by Thermal Grating Technique, *J. Phys. Chem. C* **123**, 14914–14920; 10.1021/acs.jpcc.9b02288 (2019)), which is significantly smaller than the off time between laser pulses used in our experiment ($\sim 333 \mu\text{s}$). Consequently, we believe that while heating of the sample can be a concern for higher rep. rate experiments and high fluences, it is of less concern in our work.

In the revised supplementary information, we now show in section 3, Figure 3 that the TA signal increases linearly with the carrier density across the entire excitation fluence regime to evidence that no high-density saturation was observed. We replaced the '200 ps carrier lifetime' by 'heat transport time to the substrate ($<9 \mu\text{s}$)'. Accordingly, we rephrased the discussion of the heat accumulation effect in the revised manuscript.

Reviewer #2 (Remarks to the Author):

The authors have done an admirable job of revising their manuscript based on the comments and critiques of the reviewers. Notably they have nearly completely revised their interpretation of the bandgap narrowing and transient frequency dependent photocurrent. However, the provided

explanations invoking Anderson localization + hopping, and Mahan excitons seem similarly ad hoc as their previous explanations, and inconsistent with the larger body of work of the lead halides. For example, if carriers were localized and transport was dominated by hopping, the variable range hopping model of Mott predicts a temperature dependence inconsistent with that observed experimentally (exponential rather than the observed power law). The $T^{-3/2}$ is the naive expectation from acoustic phonon scattering, so why this is not discussed or ruled out? Mahan excitons have only recently and found to be very sensitive to the exciton binding energy (observed in MAPbBr₃ but expected in MAPbI₃ Ref 60) so if the authors are claiming their importance here, they ought to have additional evidence beyond correlative speculation. If such extrapolations beyond what their data is able to furnish are tamed, I would be in favor of publication.

Authors' response: We thank again the reviewer for his/her encouraging comments on our work. The statement 'Mahan excitons have only recently (been observed) and (were) found to be very sensitive to the exciton binding energy', is correct for MAPbBr₃, for which experimental evidence has been reported. However, such experimental evidence is yet to be shown for MAPbI₃ (see also ref. 60). The authors of ref. 60 have not checked for a corresponding excitonic enhancement of the excited states in MAPbI₃. While our observation of weak excitonic screening in MAPbI₃ is quite similar to the findings reported for MAPbBr₃, we cannot claim that Mahan exciton formation is indeed evidenced in our work. In fact, the anomalous BGR found by us and reported in our work is primarily due to the linear enhancement of the TA signal in the high energy region. The type of exciton is less important, in other words, they do not have to be Mahan excitons. The Anderson-localization model suggests a coexistence of band-like and hopping transport, and thus can result in a carrier mobility roughly obeying the $T^{-3/2}$ dependence (see ref. 24 below) as well.

[24]. Lacroix, A., Laissardière, G. T. de, Quémerais, P., Julien, J.-P. & Mayou, D. Modeling of Electronic Mobilities in Halide Perovskites: Adiabatic Quantum Localization Scenario. *Phys. Rev. Lett.* **124**, 196601; 10.1103/PhysRevLett.124.196601 (2020).

Since the evidence for Mahan exciton formation presented is not sufficient in our work, we rephrased the discussion of the Mahan-exciton scenario in the revised manuscript, as suggested by the reviewer, to avoid any over-interpretation of our data leading to possibly questionable conclusions.

We thank the reviewer for pointing out that the Anderson-localization scenario (i.e. the combination of bandlike + hopping-type carrier transport) is inconsistent with the large body of work on lead halide perovskites. We note that models which use standard bandlike transport such as the Drude-Debye model, Drude-Smith model, and Drude-Lorentz model all fall short to explain our data in one way or another. Fits to our data using any of the aforementioned models indicate either a photo-induced enhancement of the dielectric response or a photo-induced carrier localization. To some extent both are related to photo-induced lattice fluctuations, which is one of the main conclusion of our work. As suggested by the reviewer, we rephrased our discussion on the Anderson-localization scenario in the revised manuscript to exclude any possible contradicting statements.

Reviewer #3 (Remarks to the Author):

Wang et al have diligently addressed the numerous scientific and technical issues raised in my previous review, and I now find this work to be highly convincing and comprehensive and publishable in Nature Communications. Congratulations on some very nice work!

While I do not require further review of the manuscript, I would strongly recommend that the authors discuss the comparison of photophysics in triple-cation MHPs (studied in the current manuscript) with other reported conventional MHPs to draw broader conclusions. Also, the referencing style is not consistent, which should be corrected.

Authors' response: We thank Dr. Ajay Jha for his encouraging comments on our work. We agree with him that the comparison of the photophysics of multi-cation mixed halide perovskites with conventional metal-halide perovskites could allow to draw broader conclusions. We are also curious to see the density-dependent BGR and the frequency-dependent THz spectra for other members of the perovskite family. However, this requires a significant amount of additional experimental work and data analysis, and we believe that, while it would be nice to have such a comparison, the main focus of our present work is on processes in perovskite materials that are considered current state-of-the-art perovskites used in single-junction solar cells ($E_g \sim 1.6$ eV, conventional MA, double-cation FAMA, triple-cation CsFAMA).

We thank the reviewer for bringing to our attention that the referencing style is not consistent. This has been corrected in the revised manuscript.

REVIEWERS' COMMENTS

Reviewer #1 (Remarks to the Author):

The authors have addressed all my concerns and I can support publication.

Reviewer #2 (Remarks to the Author):

With the current revisions, the manuscript is fit for publication.